# ON THE ROLE OF TEMPERATURE SAMPLING IN TEST-TIME SCALING

## ABSTRACT

Large language models (LLMs) can improve reasoning at inference time through test-time scaling (TTS), where multiple reasoning traces are generated and the best one is selected. Prior work shows that increasing the number of samples $K$ steadily improves accuracy. In this paper, we demonstrate that this trend does not hold indefinitely: at large $K$, further scaling yields no gains, and certain *hard* questions remain unsolved regardless of the number of traces. Interestingly, we find that different sampling temperatures solve different subsets of problems, meaning single-temperature scaling explores only part of a model's potential. We therefore propose scaling along the temperature dimension, which enlarges the reasoning boundary of LLMs. Temperature scaling enables base models to reach performance comparable to reinforcement learning (RL)-trained counterparts, without additional post-training. We further provide a comprehensive analysis of this phenomenon and design a multi-temperature voting method that reduces the overhead of temperature scaling. Overall, our findings suggest that TTS is more powerful than previously thought, and that temperature scaling offers a simple and effective way to unlock the latent potential of base models.

## 1 INTRODUCTION

Large language models (LLMs) have demonstrated strong reasoning capabilities for complex problems at test time (Wei et al., 2022). As illustrated in Figure 1a, two main approaches have emerged to achieve such reasoning. The first trains models to produce long reasoning traces with self-reflection and correction, often implemented through reinforcement learning (RL) (Guo et al., 2025a; Yang et al., 2025c). While effective, this approach requires costly and time-consuming training (Liu et al., 2025a). The second, known as test-time scaling (TTS) (Brown et al., 2024; Snell et al., 2025; Zhao et al., 2025), shifts the burden to inference: the model generates multiple reasoning traces in parallel and a verifier selects the most reliable one (Saad-Falcon et al., 2025a). Unlike RL, TTS requires no large-scale post-training and generates relatively short, prefix-sharing traces. This enables efficient reuse of the KV cache (Hooper et al., 2025) and offers speed advantages in modern serving systems (Kwon et al., 2023; Zheng et al., 2024).

Recent studies on TTS have shown that increasing the number of samples $K$ can enhance reasoning performance (Schaeffer et al., 2025). As illustrated in Figure 1b, scaling $K$ up to 1,024 shows a clear trend of steadily improving accuracy. However, when we push $K$ further to 13,312, the improvement stops: some questions remain unsolved no matter how many samples are drawn. *If further scaling $K$ brings no improvement, have we reached the ceiling of TTS performance?*

The answer is no. As shown in Figure 1c, we observe an interesting phenomenon: when scaling $K$ up to 1,024 under different sampling temperatures $T$, the sets of solvable questions differ. For example, a question unsolved at $T = 0.5$ may become solvable at $T = 0.7$. The model's overall solvable set is therefore larger than the solvable set under any single temperature. This indicates that scaling $K$ at a fixed temperature only explores part of the model's potential. To unlock the full boundary, we scale along the temperature dimension: given any budget, we divide samples evenly across multiple temperatures. As shown in Figure 1d, this multi-temperature strategy achieves a much higher performance upper bound. These results highlight the importance of temperature sampling in TTS.

In this paper, we show that temperature provides a new dimension for scaling at test time. Through extensive experiments across model sizes and datasets, we demonstrate that temperature scaling ex-

Figure 1: Observations and motivation for temperature scaling in TTS. (a) RL vs. TTS: RL produces long single traces, while TTS generates multiple shorter ones. (b) Pass@$K$ and $-\log($Pass@$K)$ curves at $T = 0.7$ on Qwen3-4B (AIME 2025); no gain beyond $K = 1{,}024$. (c) Question solvability on AIME 2025 for Qwen3-4B: different temperatures solve different subsets of questions. (d) Single-temperature vs. multi-temperature scaling: the latter expands the reasoning boundary.

pands the reasoning boundary of LLMs. This effect arises because, while all temperatures can solve *easy* questions, some *hard* questions are only solvable at specific temperatures. With temperature scaling, a base model can reach performance comparable to RL-trained models. We further conduct entropy-based analyses and case studies to understand this phenomenon. Finally, we design a multi-temperature voting method that identifies and exits *easy* questions early, making temperature scaling more efficient. Overall, our contributions are threefold:

- **Scaling test-time compute to a new dimension.** We show that scaling the sampling temperature expands the model's reasoning boundary, revealing a new axis of test-time compute.
- **Analyzing the dynamics of temperature scaling.** Through comprehensive analysis, we show that the enlarged reasoning boundary arises because different temperatures solve different *hard* questions, while *easy* questions can be solved by all temperatures.
- **Designing efficient methods for temperature scaling.** We propose a multi-temperature voting strategy that exits *easy* questions early, reducing overhead while preserving the benefits of temperature scaling across models and datasets.

## 2 SCALING TEMPERATURE AT TEST TIME

In this section, we first introduce the concept of temperature sampling and describe the experimental setup for scaling temperature. We then present the performance improvements achieved through temperature scaling, and compare this approach against further scaling $K$ and RL-based methods.

### 2.1 TEMPERATURE SAMPLING

**Temperature sampling.** Temperature-based sampling has long been studied in probabilistic modeling (Ackley et al., 1985), and it remains a core component of decoding in LLMs. At each step of autoregressive generation, the model conditions on the input $x$ and previously generated tokens $y_{1:t-1}$ to yield a distribution over the next token, from which $y_t$ is sampled:

$$y_t \sim p(\cdot \mid x, y_{1:t-1}) = \mathrm{softmax}\left(\frac{f_\theta(x, y_{1:t-1})}{T}\right).$$

Here, $f_\theta(\cdot)$ denotes the model logits and $T$ is the sampling temperature. The temperature rescales the logits before applying the softmax, thereby shaping the probability distribution used for sampling.

**Effect of temperature.** The sampling temperature $T$ is non-negative. When $T = 0$, the distribution collapses to a delta on the maximum-logit token, equivalent to deterministic decoding. For $T > 0$, smaller values sharpen the distribution, making generation more deterministic and favoring high-probability tokens, while larger values flatten it, increasing randomness and promoting diversity.

### 2.2 EXPERIMENTAL SETUP

**Sampling temperature and number of samples.** We vary the sampling temperature from 0.0 to 1.2 in increments of 0.1. At temperature 0.0, we generate a single reasoning trace for each question, whereas at other temperatures we generate 1,024 traces per question.

Table 1: Results (%) of temperature scaling across models and datasets. Base reports Pass@1,024 under $T = 0.6$. $+T$ reports Pass@All when scaling the sampling temperature from 0.0 to 1.2, with 1,024 samples per temperature. $\Delta$ denotes the difference between $+T$ and Base.

| Models | AIME2025 | | | AIME2024 | | | MATH500 | | | LiveCodeBench | | | Hi-ToM | | |
|---|---|---|---|---|---|---|---|---|---|---|---|---|---|---|---|
| | Base | $+T$ | $\Delta$ | Base | $+T$ | $\Delta$ | Base | $+T$ | $\Delta$ | Base | $+T$ | $\Delta$ | Base | $+T$ | $\Delta$ |
| Qwen3-8B | 60.0 | 66.7 | **6.7** | 73.3 | 80.0 | **6.7** | 97.0 | 97.8 | **0.8** | 32.6 | 40.0 | **7.4** | 93.0 | 95.5 | **2.5** |
| Qwen3-4B | 60.0 | 73.3 | **13.3** | 66.7 | 76.7 | **10.0** | 95.5 | 98.5 | **3.0** | 36.0 | 40.0 | **4.0** | 83.0 | 92.0 | **9.0** |
| Qwen3-1.7B | 46.7 | 50.0 | **3.3** | 43.3 | 50.0 | **6.7** | 92.5 | 95.5 | **3.0** | 29.7 | 35.4 | **5.7** | 37.0 | 55.0 | **18.0** |
| Qwen3-0.6B | 20.0 | 36.7 | **16.7** | 23.3 | 30.0 | **6.7** | 88.1 | 94.0 | **5.9** | 25.7 | 31.4 | **5.7** | 71.5 | 82.5 | **11.0** |

**Datasets and Prompts.** We evaluate on reasoning benchmarks spanning multiple domains. AIME 2024, AIME 2025, and MATH500 (Hendrycks et al., 2021) are used to assess mathematical reasoning. LiveCodeBench v6 (Jain et al., 2025) evaluates code generation, and Hi-ToM (Wu et al., 2023) focuses on social and logical reasoning. All these benchmarks support automatic output verification. Further details about the datasets and prompts are provided in Appendix A.

**Platform and models.** All experiments are conducted using vLLM (Kwon et al., 2023) to enable large-batch rollouts. We run our evaluations on a cluster using 64 NVIDIA H100 GPUs. The models include the Qwen3 series (0.6B, 1.7B, 4B, 8B) (Yang et al., 2025a), and Polaris-4B-Preview (An et al., 2025), an RL-trained variant of Qwen3-4B.

**Evaluation metrics.** Our primary evaluation metric is Pass@$K$, where $K$ is the number of sampled traces. It measures the probability of obtaining at least one correct answer when sampling $K$ times. Let $N$ be the total number of generated samples and $C$ the number of correct ones, then

$$\text{Pass@}K = 1 - \frac{\binom{N-C}{K}}{\binom{N}{K}}.$$

In addition, the average accuracy Avg@$N = C/N$ approximates the model's correctness probability when $N$ is large. As our aim is to highlight scaling behavior, we use ground-truth verification instead of a reward model (RM) to avoid confounds from RM quality.

**AIME 2024/2025 reasoning trace validation.** Each AIME problem has an integer answer. It has been argued that LLMs may sometimes arrive at the correct answer by chance rather than through valid reasoning (Wen et al., 2025). To address this concern, we conduct additional validation on all problems where a model's Avg@1,024 is below 3%. For these problems, the model-generated reasoning traces and human-written reference solutions are jointly reviewed by gpt-5 as an automatic judge. Further details of this validation process are provided in Appendix A.

## 2.3 RESULTS ON TEMPERATURE SCALING

**Main results.** We compare scaling at the default temperature ($T = 0.6$) with multi-temperature scaling. As shown in Table 1, temperature scaling consistently enlarges the reasoning boundary across all models and datasets. For example, on AIME 2025, Qwen3-4B gains 13.3 points after temperature scaling, indicating that any single temperature covers only part of the model's reasoning capability. However, the effect varies by dataset: on MATH500, where Qwen3-8B already performs strongly, the additional gain from temperature scaling is relatively small.

**No single temperature works for all questions.** Each question shows its own temperature preference. For example, in Figure 2a, one AIME 2025 problem requires $T = 1.1$ with 128 traces to reach Pass@$K = 1$, while $T = 0.7$ needs 256 traces. Figure 2b further shows the temperature preferences of Qwen3-4B on all AIME 2024/2025 questions: each question has a different optimal temperature, and no single value performs best across the board. Interestingly, while higher temperatures increase diversity and creativity, we find no consistent link to improved reasoning capability.

**Pass@$K$ vs. Avg@$N$.** Figure 2c shows Pass@$K$ scaling curves of Qwen3-4B on AIME2025 under different temperatures. When $K$ is small, different temperatures yield similar Pass@$K$, since models mainly solve *easy* questions that all temperatures can handle. As $K$ grows, however, temperature-specific advantages emerge: a *hard* question favoring one temperature may eventually be solved there but not at others, leading to larger gaps at Pass@1,024. In contrast, Figure 2d shows that

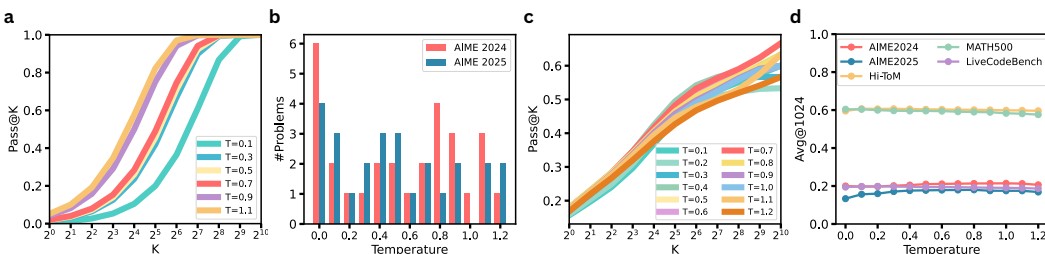

Figure 2: Scaling temperature for test-time compute on Qwen3-4B. (a) Pass@$K$ curves for different temperatures on AIME 2025 Q22. (b) Distribution of preferred temperatures across AIME 2024/2025. (c) Pass@$K$ scaling curves on AIME 2025. (d) Avg@$1,024$ curves across five datasets.

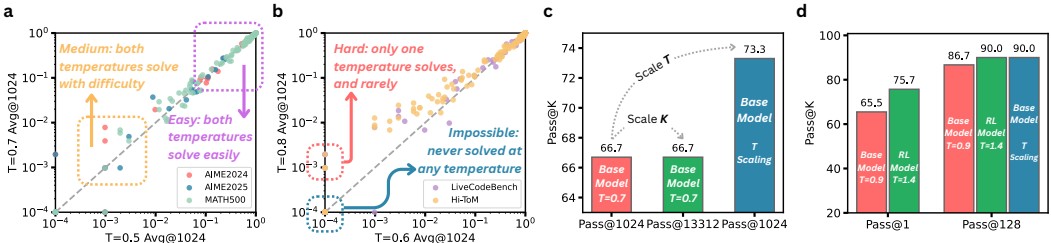

Figure 3: Comparison of scaling along $K$ and scaling along $T$. (a) Correlation of Avg@$1,024$ across two temperatures on AIME2025, Qwen3-4B. (b) Correlation of Avg@$1,024$ across two temperatures on AIME2025, Qwen3-8B. (c) Scaling $K$ vs. scaling $T$ on AIME2025, Qwen3-4B. (d) Temperature scaling vs. RL-trained model on AIME2025, Qwen3-4B, thinking mode.

Avg@$1,024$ remains nearly identical across temperatures, as most *easy* questions are solved by all settings and the ability to solve harder ones is drowned out in this metric.

**Remarks.** Prior work has reported similar Avg@$N$ curves and argued that, within a range, temperature does not affect LLM reasoning performance (Renze, 2024). Our results suggest this is not the case in the TTS setting: Avg@$N$ fails to capture the model's ability to solve *hard* questions with low probability, whereas a good verifier can still identify these sparse correct traces (Zhao et al., 2025).

## 2.4 SCALING ALONG $T$ VS. SCALING ALONG $K$

**Understanding dataset difficulty distribution and solvability.** Figures 3a and 3b plot the Avg@$1,024$ of each question under two different temperatures, with each point representing one problem. From this view, the questions can be grouped as:

- *Easy*: questions near the upper-right diagonal, solved by both temperatures with high probability.
- *Medium*: points near the lower-left diagonal, harder but still solved by both temperatures.
- *Hard*: points lying on the axes, solvable at one temperature with low probability but not the other.
- *Impossible*: points at the origin, unsolved regardless of temperature.

We observe that many questions cluster along the upper-right diagonal: these are simple problems that any temperature can solve, explaining why Avg@$N$ shows little difference across temperatures. Some questions remain at the origin: these are beyond the reach of temperature scaling and likely require training rather than TTS.

**Scaling across temperatures unlocks latent capability.** The axis points (*hard* problems) illustrate why temperature scaling is effective: sampling across multiple temperatures ensures that these problems enter the solvable set, whereas sampling under a single temperature collapses all solvable *hard* problems from other temperatures back to the origin.

**Further scaling the number of samples does not help.** As shown in Figure 1b, increasing $K$ from 1,024 to 13,312 does not solve any additional problems. This suggests that the TTS curve has two

phases: an initial regime where more samples improve performance, followed by a plateau where no further gains are observed. In contrast, Figure 3c shows that scaling temperature yields a 6.67% improvement, demonstrating that temperature provides a meaningful additional dimension for TTS.

**Remarks.** When scaling $K$ to 13,312, we initially observe improvements, with the model producing correct final answers for questions unsolved under a smaller budget. However, after gpt-5-based trace verification, these cases are found to be invalid: the model is merely more likely to guess the correct answer with more samples. This highlights that future work on RL and TTS should account not only for final-answer accuracy but also for the validity of reasoning traces.

### 2.5 Comparison with RL-Trained Methods

**Setup.** We evaluate the RL-trained Polaris-4B-Preview model against its base model, Qwen3-4B. Following the setup suggested by An et al. (2025), we run the RL model on AIME 2025 with temperature $T = 1.4$, a maximum context length of 96K tokens, and the "thinking" mode enabled. To ensure fairness, we apply the same configuration to the base model: scaling temperature from 0.0 to 1.4, enabling "thinking" mode, and setting $N = 128$ samples per question.

**Scaling $K$ cannot make the base model comparable to RL.** As shown in Figure 3d, the RL-trained model consistently outperforms the base model when scaling the number of samples, with a clear advantage under small budgets (e.g., Pass@1). Although the performance gap narrows as $K$ increases, the base model never fully catches up.

**Further scaling $T$ can make the base model comparable to RL.** As shown in Figure 3d, scaling across temperatures raises the base model to Pass@All performance on par with the RL-trained model. The overall success rates are similar, though they differ in which questions remain unsolved: the base model fails on Q14, Q15, and Q28, while the RL model fails on Q13, Q14, and Q15. Thus, temperature scaling extends the base model's reasoning boundary to match RL performance.

**Remarks.** There is an active debate on whether RL brings genuinely new capabilities to LLMs (Liu et al., 2025a). Some prior work argues that scaling $K$ sufficiently can erase or even reverse the advantage of RL (Yue et al., 2025), while others contend that many of the base model's large-$K$ successes are merely guesses, and after verifier filtering the RL-trained model remains stronger (Wen et al., 2025). Our findings align with the latter: scaling $K$ narrows the gap but does not eliminate it. However, when we further scale across $T$, the base model becomes comparable to RL. In this sense, exploring both $K$ and $T$ dimensions reveals a broader reasoning boundary than either alone.

## 3 Deep Analysis of Temperature Scaling

In this section, we first present the entropy dynamics underlying temperature scaling, then provide a case study to illustrate these behaviors in practice, and finally discuss the strengths and limitations of temperature scaling.

### 3.1 Learning the Entropy Dynamics of Temperature Scaling

**Entropy.** We measure model uncertainty using the entropy of the next-token distribution. At each step, given logits $f_\theta(x, y_{1:t-1})$, we compute

$$H = -\sum_y p(y \mid x, y_{1:t-1}) \log p(y \mid x, y_{1:t-1}), \quad p(y \mid x, y_{1:t-1}) = \text{softmax}(f_\theta(x, y_{1:t-1})).$$

We report the average entropy $H$ across all generated tokens. Low $H$ indicates sharp logits and high confidence, while high $H$ reflects uncertainty. Here, it is always computed from the model's untempered softmax. As shown in Figure 4a–c, average $H$ rises with $T$: higher $T$ increases the chance of selecting low-probability tokens, raising the sequence-level $H$.

**Entropy dynamics of correct vs. incorrect traces.** We analyze the average entropy of all traces, as well as the correct and incorrect subsets. For *easy/medium* questions (Figure 4a), correct-trace entropy increases smoothly with temperature, while incorrect-trace entropy rises much faster. At low $T$, correct traces show higher entropy, reflecting greater diversity; at high $T$, they show lower entropy, reflecting coherence when the model answers correctly. For *impossible* questions (Figure 4c),

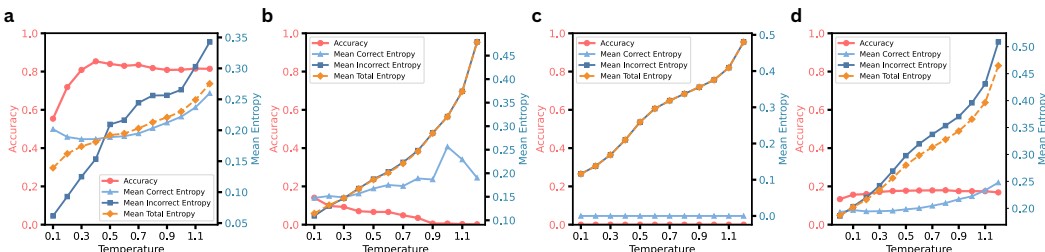

Figure 4: Entropy dynamics of Qwen3-4B across temperatures on AIME 2025. (a) Q16. (b) Q29. (c) Q11. (d) Across the whole dataset.

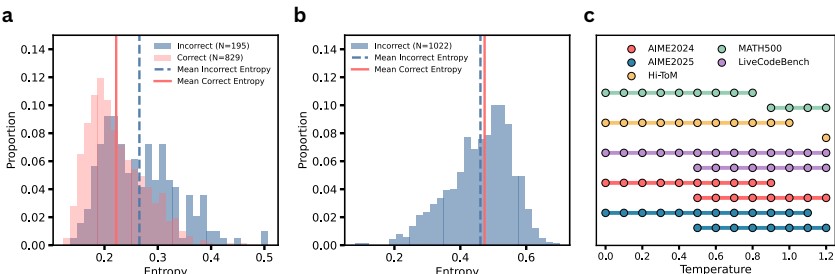

Figure 5: Entropy distributions and temperature subset. (a) An *easy* problem (AIME 2025 Q16, Qwen3-4B). (b) A *hard* problem (AIME 2025 Q25, Qwen3-4B). (c) Temperature minimal subset for Qwen3-4B.

in contrast, the entropy of all traces increases rapidly as $T$ grows. At the dataset level (Figure 4d), correct traces consistently maintain lower entropy than incorrect ones at higher temperatures, suggesting that *the model **seems** to "know" when it is answering correctly*. But is this always the case?

**When does the model "know it knows"?** The entropy gap between correct and incorrect traces holds reliably only for *easy/medium* questions. As shown in Figure 5a, for an *easy* problem at $T = 1.0$, the entropy distribution of correct traces is clearly shifted to the left compared to incorrect ones. However, for a *hard* problem (Figure 5b), where only 2 out of 1,024 traces are correct, the entropy of correct traces is not always lower than that of incorrect ones. The model "knows it knows" only when the question is *easy*, but not when it is *hard*. This explains why in Figure 4d, where *easy* problems dominate, the aggregate effect makes it appear that correct traces always have lower entropy, drowning out the behavior on *hard* problems.

**Remarks.** Recent work has explored uncertainty-guided decoding (Kadavath et al., 2022; Kang et al., 2025), where high-entropy traces are discarded under the assumption they are unlikely correct (Fu et al., 2025). Our findings suggest this assumption holds for *easy/medium* questions, where correct traces typically align with lower entropy. However, for *hard* questions, correct traces do not necessarily exhibit low entropy, indicating uncertainty alone is insufficient as a universal signal.

## 3.2 CASE STUDY

**Problem.** We illustrate the effect of sampling temperature using AIME 2025 Q24. The task is to determine the number of zeros of the function $f(x) = \sin(7\pi \sin(5x)), 0 < x < 2\pi$. This problem essentially has a single correct reasoning path: one must reduce the condition $f(x) = 0$ to $\sin(5x) = k/7$ with $k \in \{-7, \ldots, 7\}$, and count the corresponding solutions over five periods.

**Case Study.** At temperature $T = 0.7$, the model solved the problem in 2 out of 1,024 samples. The successful traces explicitly reduced the condition to $\sin(5x) = k/7$, enumerated the valid $k$ values, and counted their solutions within $(0, 2\pi)$. In contrast, at $T = 0.9$, none of the sampled traces followed this reasoning structure; typical attempts instead guessed the number of zeros by listing approximate intersection points, without connecting them to the $k/7$ values, and thus failed to reach

Figure 6: Overview of the voting algorithm for efficient temperature scaling. Each temperature maintains its own candidate pool and first performs intra-temperature voting with threshold $\tau_{\text{intra}} = 0.8$. Only if all temperatures pass this stage do we proceed to cross-temperature voting, where majority votes are aggregated across temperatures with threshold $\tau_{\text{cross}} = 1.0$. Questions that meet both criteria are marked as *easy* and exit early.

the correct count. This illustrates that a *hard* question may be solvable under one temperature but not another, highlighting the need for temperature scaling.

### 3.3 STRENGTHS AND LIMITATIONS OF TEMPERATURE SCALING

**Strengths.** As shown in Figure 1d, the most notable strength of temperature scaling is that it achieves a higher upper bound when $K$ is large, revealing a larger reasoning boundary. A second advantage is that distributing samples across temperatures does not substantially deviate from any single-temperature Pass@$K$ curve when $K$ is small. This is because improvements at small budgets mainly come from solving *easy* questions, for which accuracy is largely unaffected by the choice of temperature (as also shown in Figures 2d and 3a,b).

**Limitations.** The main limitation of temperature scaling lies in computational efficiency. For example, using 12 temperatures requires roughly $12\times$ the compute of single-temperature scaling to reach the expanded reasoning boundary. This overhead is less of a concern when a strong verifier is available: most *easy* questions can be solved with little computation and verified early, allowing the budget to be focused on *hard* ones. However, such verifiers are not always accessible. In the next section, we investigate whether some of this cost can be saved even *without* verifiers.

## 4 DESIGN TEST-TIME METHODS WITH TEMPERATURE SCALING

In this section, we first analyze whether all temperatures and all questions are necessary for temperature scaling. We then design an algorithm for more efficient temperature sampling.

### 4.1 ARE ALL TEMPERATURES AND QUESTIONS NECESSARY FOR SCALING?

**Finding a minimal subset of temperatures.** A natural question is whether we must sample from all available temperatures to achieve the expanded upper bound, or whether a smaller subset is sufficient. As shown in Figure 5c, we evaluate subsets formed by gradually adding temperatures from either the low or high end. The results show that starting from higher temperatures requires a smaller subset to reach the upper bound; traces generated at low temperatures can also be obtained at higher ones. Based on this, we select a subset that generalizes across models and datasets, excluding very low temperatures (0.1–0.3).

**Early exit for *easy* questions under temperature scaling.** Figures 3a and 3b show that *easy* questions require far fewer samples to solve and can be answered reliably at any temperature with high probability. This observation motivates a simple strategy: avoid redundant sampling on such problems by using a voting-based mechanism across temperatures. In the following, we describe how multi-temperature voting can serve as an early-exit method.

### 4.2 ALGORITHM FOR EFFICIENT TEMPERATURE SCALING

**Method overview.** As illustrated in Figure 6, for each question we maintain per-temperature candidate answer pools. In each round, every temperature contributes one new trace per active question,

Table 2: Results across five datasets. Base reports Pass@1,024 at $T = 0.6$. $+T$ reports results with full temperature scaling. $+Ours$ denotes our efficient temperature-scaling method. $\Delta C$ shows the reduction in computation cost relative to full temperature scaling.

| Models | AIME2025 | | | | AIME2024 | | | | MATH500 | | | |
|---|---|---|---|---|---|---|---|---|---|---|---|---|
| | Base | $+T$ | $+Ours$ | $\Delta C$ | Base | $+T$ | $+Ours$ | $\Delta C$ | Base | $+T$ | $+Ours$ | $\Delta C$ |
| Qwen3-8B | 60.0 | 66.7 | 66.7 | **-31.6%** | 73.3 | 80.0 | 80.0 | **-31.6%** | 97.0 | 97.8 | 97.8 | **-54.4%** |
| Qwen3-4B | 60.0 | 73.3 | 73.3 | **-33.5%** | 66.7 | 76.7 | 76.7 | **-31.6%** | 95.5 | 98.5 | 98.5 | **-49.5%** |
| Qwen3-1.7B | 46.7 | 50.0 | 50.0 | **-27.2%** | 43.3 | 50.0 | 50.0 | **-27.2%** | 92.5 | 95.5 | 95.5 | **-36.3%** |
| Qwen3-0.6B | 20.0 | 36.7 | 36.7 | **-25.0%** | 23.3 | 30.0 | 30.0 | **-25.0%** | 88.1 | 94.0 | 94.0 | **-26.0%** |

| Models | LiveCodeBench | | | | Hi-ToM | | | | Average | | | |
|---|---|---|---|---|---|---|---|---|---|---|---|---|
| | Base | $+T$ | $+Ours$ | $\Delta C$ | Base | $+T$ | $+Ours$ | $\Delta C$ | Base | $+T$ | $+Ours$ | $\Delta C$ |
| Qwen3-8B | 32.6 | 40.0 | 40.0 | **-35.0%** | 93.0 | 95.5 | 94.5 | **-78.7%** | 71.2 | 76.0 | 75.8 | **-46.3%** |
| Qwen3-4B | 36.0 | 40.0 | 40.0 | **-32.8%** | 83.0 | 92.0 | 86.5 | **-32.8%** | 68.2 | 76.1 | 75.0 | **-36.0%** |
| Qwen3-1.7B | 29.7 | 35.4 | 35.4 | **-29.9%** | 37.0 | 55.0 | 42.0 | **-78.3%** | 49.8 | 57.2 | 54.6 | **-39.8%** |
| Qwen3-0.6B | 25.7 | 31.4 | 31.4 | **-26.1%** | 71.5 | 82.5 | 81.5 | **-32.4%** | 45.7 | 54.9 | 54.7 | **-26.9%** |

and the corresponding answers are recorded. We first perform *intra-temperature* voting: for each temperature, the most frequent answer is selected, and if its vote count does not reach the intra-threshold $\tau_{\text{intra}}$, that temperature is deemed not yet confident and sampling continues. Only when all temperatures satisfy the intra-threshold do we proceed to *cross-temperature* voting, where the majority answers from each temperature are voted across temperatures. If the winning answer reaches the cross-threshold $\tau_{\text{cross}}$, the question is marked as *easy* and exits early. Otherwise, sampling continues in the next round. The complete algorithm is provided in Appendix C.

**Remarks.** It is well known that the answer most favored by the model is not always correct (Brown et al., 2024). We acknowledge this limitation. However, our goal here is not to aggregate answers for final prediction, but to identify which questions are *easy* and can exit early. Moreover, the use of multi-temperature voting helps smooth out spurious signals, and the relatively high thresholds we adopt ($\tau_{\text{intra}} = 0.8$, $\tau_{\text{cross}} = 1.0$) provide additional robustness against noise.

## 5 EXPERIMENTS

In this section, we evaluate the proposed method for more efficient temperature sampling. We then analyze the results and discuss directions for future work.

### 5.1 RESULTS AND ANALYSIS

**Main results.** As shown in Table 2, our method reduces the computation required for temperature scaling across tasks, while maintaining nearly the same performance. The efficiency gains come from excluding very low temperatures and enabling early exit on *easy* questions. For example, on MATH500, Qwen3-8B achieves a 54.4% reduction in computation with negligible loss in Pass@All. Notably, Hi-ToM shows a different pattern: as a belief-allocation reasoning task, it can be solved by powerful models but also occasionally by weaker ones for spurious reasons, leading to less consistent scaling behavior.

**A powerful model is what we need.** As shown in Table 2, the overall computation savings also follow a scaling trend. A strong yet compact model, such as Qwen3-8B, is particularly well-suited for temperature scaling: it classifies more questions as *easy*, enabling our method to evict them early for efficiency and apply temperature scaling only to the *hard* ones. By contrast, models that are too small struggle even on *easy* questions, leaving little room for efficient scaling.

### 5.2 DISCUSSION

**Alternative strategies for evicting *easy* questions.** Beyond multi-temperature voting, we also explored using entropy signals to identify questions that can be exited early. If the entropy remains very low and changes little across temperatures, the question is likely *easy*; if the entropy is very high, the question is likely *impossible*, and repeated sampling will not help. However, we find that

entropy distributions vary significantly across datasets and domains, making it difficult to design a universal entropy-based eviction strategy.

**Temperature sampling in feedback-based workflows.** In this paper, we focus on internal model signals to evict *easy* questions. In real applications, this is less of a concern: one can simply call a powerful verifier after each run, and if a trace is deemed correct, the question can be evicted immediately. This would allow temperature sampling to focus only on unsolved questions, yielding substantial savings. However, in such workflows the verification cost itself becomes a critical factor. Regardless, we believe temperature sampling remains a simple yet powerful way to explore the reasoning boundary of base models. An interesting direction for future work is to study how variable temperature within a single trace may further influence reasoning ability.

## 6 RELATED WORK

**TTS for LLMs.** Multi-trace TTS (Brown et al., 2024; Snell et al., 2025; Schaeffer et al., 2025; Zhao et al., 2025; Liu et al., 2025b; Khairi et al., 2025) generates multiple candidate completions in parallel and selects the best one using either a verifier (Wang et al., 2024; Gou et al., 2024; Sun et al., 2024; 2025; Singh et al., 2025; Guo et al., 2025b; Dorner et al., 2025; Saad-Falcon et al., 2025a) or voting-based approaches (Wang et al., 2023; 2025c). This approach can be further combined with search algorithms (Yao et al., 2023; Bi et al., 2025; Chatziveroglou, 2025; Qiu et al., 2024; Zhang et al., 2024a; Lin et al., 2025), which interleave generation and selection in a step-by-step manner to progressively improve the final output.

Recent studies have also explored the connection between RL and TTS (Sareen et al., 2025; Zuo et al., 2025; Yue et al., 2025; Wen et al., 2025; Liu et al., 2025a). In parallel, other works investigate how to teach models to dynamically choose between single-trace and multi-trace reasoning strategies (Yang et al., 2025b; Biju et al., 2025; Jin et al., 2025; Wang et al., 2025f). Recent work also incorporates TTS into reasoning workflows (Saad-Falcon et al., 2025b; Chakraborty et al., 2025; Shen et al., 2025).

**Sampling Methods for TTS.** A typical sampling pipeline consists of three components: the prompt input to the model, the stochastic sampling procedure, and post-processing of the logits. Prior work (Wang et al., 2025b; Liu et al., 2025c) has shown that the design of the prompt can influence the inference performance. Based on temperature sampling (Ackley et al., 1985), many works have explored new strategies for sampling (Meister et al., 2023; Li et al., 2024a; Zhu et al., 2024b; Zhang et al., 2024b; Cai et al., 2024; Dhuliawala et al., 2024; Li et al., 2025b) and logit truncation (Fan et al., 2018; Holtzman et al., 2020; Basu et al., 2021; Hewitt et al., 2022; Minh et al., 2025) to better balance diversity and quality. Recently, Renze (2024) reported that temperature has limited impact on model reasoning capability. Our results suggest this is not the case when using Pass@$K$ as the metric. Meanwhile, Du et al. (2025) found that the optimal temperature varies across datasets, a pattern we also observe in Figure 2d.

More related work on single-trace TTS, efficient algorithms, and serving systems can be found in Appendix D.

## 7 CONCLUSION

We revisit TTS for reasoning in LLMs and show that scaling the number of samples $K$ alone has diminishing returns: once $K$ is large, accuracy plateaus and some questions remain unsolved. In contrast, scaling the sampling temperature expands the reasoning boundary, as different temperatures unlock different *hard* problems while *easy* ones are solved universally. This makes temperature scaling a simple yet powerful complement to traditional TTS, enabling base models to match RL-trained counterparts without additional training. Through entropy-based analysis and case studies, we further characterize the dynamics behind this effect. Finally, we propose efficient multi-temperature voting methods that cut the overhead of temperature scaling by exiting early on *easy* questions. Overall, our results highlight temperature scaling as an effective and practical tool to unlock the latent reasoning capabilities of LLMs.

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

# APPENDIX

## A  DATASETS, EVALUATION, AND VERIFICATION

### A.1  DATASETS AND EVALUATION METHODS

**Mathematical reasoning.** We use AIME 2024, AIME 2025, and MATH500 (Hendrycks et al., 2021) to evaluate mathematical reasoning. AIME is a high school mathematics competition that features 30 challenging problems each year, and we include all 30 problems from both the 2024 and 2025 editions. MATH500 is a subset of the MATH benchmark comprising challenging problems across multiple topics; we select all problems of difficulty level 5, which is the highest level in MATH500, yielding 134 questions. For evaluation, we prompt the model to place its final answer inside a \boxed{} expression. For AIME, where answers are integers, the boxed content can be directly parsed and compared to the reference. For MATH500, where boxed expressions can be more complex, we use sympy to check mathematical equivalence between the predicted and reference answers.

**Code generation.** We use LiveCodeBench v6 (Jain et al., 2025), which consists of recently collected programming problems from AtCoder and LeetCode. Version 6 covers 175 problems released between January 2025 and May 2025. To evaluate model outputs, we run each generated program against a large set of private test cases, and a solution is considered correct only if it passes all test cases.

**Social and logical reasoning.** Theory-of-mind (ToM) refers to the ability to infer others' mental states such as beliefs (Sclar et al., 2025; Wu et al., 2025b). A common ToM evaluation format is the unexpected transfer task, which can be viewed as a form of commonsense dynamic logical reasoning (Wu et al., 2025c). For example, in a multi-step story: Alice and Bob are in a room with a chocolate in a box; after Alice leaves, Bob moves the chocolate to the table. A correct solver should infer that Alice still believes the chocolate is in the box. Hi-ToM (Wu et al., 2023) contains 200 problems that evaluate multi-agent ToM reasoning over long, temporally structured scenarios. To evaluate model outputs, we generate process-level labels, and a prediction is considered correct only if all beliefs are correctly inferred at every step.

### A.2  EVALUATION PROMPTS

The prompts for AIME, MATH500, and LiveCodeBench are shown in Figure 7, Figure 8, and Figure 9, respectively. For Hi-ToM, we use the same one-shot prompt format as in Wu et al. (2025c).

### A.3  VERIFICATION PIPELINE

For AIME, we apply an additional reasoning-trace verification step to reduce the risk of spurious correctness. Specifically, for every model and every AIME problem, if among 1,024 sampled traces the number of correct answers is $\leq 32$ (i.e., Avg@1024 $\leq 3\%$), we perform gpt-5-assisted validation. Gpt-5 is provided with multiple human-written reference solutions and asked to judge whether each model-generated reasoning trace is logically valid and leads to the correct answer. Only traces

```
Prompt for AIME

Please reason step by step, and put your final answer within
\boxed{}.

{Question}
```

Figure 7: Prompt used for AIME.

```
Prompt for MATH500

Answer the following math question step by step, given in LaTeX
    format, clearly and concisely, and present the final answer as
    \boxed{x}, where X is the fully simplified solution.

Example:
Question: \int_0^1 (3x^2 + 2x) \,dx
Solution: \int (3x^2 + 2x) \,dx = x^3 + x^2 + C
Evaluating from 0 to 1: (1^3 + 1^2) - (0^3 + 0^2) = 1 + 1 - 0 = 2

\boxed{2}

Now, solve the following question step by step.

{Question}
```

Figure 8: Prompt used for MATH500.

judged as correct are retained; all others are filtered out. The validation prompt is shown in Figure 10.

## B  LLM USAGE

We used GPT-5 as a general-purpose assist tool for language refinement and proofreading. In addition, GPT-5 was employed to assist in the verification of AIME reasoning traces, as described in Appendix A.

## C  ALGORITHM

The algorithm for the efficient temperature scaling can be found in Algorithm 1.

## D  EXTENDED RELATED WORK

**Single-trace TTS.** In single-trace TTS, the goal is to encourage deeper and more deliberate reasoning within a single inference path (Wei et al., 2022; Muennighoff et al., 2025). This can be achieved by RL (Guo et al., 2025a; Yang et al., 2025c; Wang et al., 2025a; Cheng et al., 2025; An et al., 2025) or by distilling reasoning traces from a stronger teacher model (Li et al., 2025a). Compared to single-trace approaches, multi-trace TTS has been shown to produce more stable results (Suvra et al., 2025; Gema et al., 2025). This work focuses on the multi-trace setting.

**Efficient Algorithms for TTS.** One line of work focuses on resource allocation, aiming to allocate more computation to difficult examples and less to easier ones (Aggarwal et al., 2023; Li et al., 2024b; Huang et al., 2025; Wu et al., 2025a; Wang et al., 2025d;e;g; Zhang et al., 2025; Fu et al.,

```
Prompt for LiveCodeBench

You are an expert Python programmer.
You will be given a programming problem and must generate a correct
    Python solution that matches the specification and passes all
    tests.

{Question}

Format:
You will use the following starter code to write the solution
and enclose your code within backticks.

```python
class Solution:
    def solve(self, ...):
        pass

Answer:
```

Figure 9: Prompt used for LiveCodeBench.

2025). Another line of work incorporates system-level techniques into TTS, such as compressing or reusing the KV cache to avoid redundant computation (Chen et al., 2025; Song et al., 2025; Ding et al., 2025; Hooper et al., 2025), or integrating with speculative decoding to reduce latency (Wang et al., 2025h; Pan et al., 2025a). These methods improve test-time efficiency by combining better search with system-level optimizations.

**Efficient Serving System for TTS.** Recent work has proposed various serving systems to reduce decoding latency. These include optimized attention kernels (Ye et al., 2025; Zadouri et al., 2025), speculative decoding methods (Miao et al., 2024), and KV cache compression techniques (Lancucki et al., 2025). Other efforts improve system throughput via techniques like continuous batching (Yu et al., 2022) and separating the prefilling and decoding stages (Qin et al., 2025).

Parallel decoding-based TTS can be viewed as a tree-structured decoding problem, where a shared prefix is expanded into multiple completions. In this setting, many queries access the same KV cache from the shared prefix, making efficient KV cache management a key challenge (Gim et al., 2024; Pan et al., 2025b; Zhu et al., 2025; Sadhukhan et al., 2025). vLLM addresses this by introducing PagedAttention (Kwon et al., 2023), which reduces memory usage and improves throughput. Building on this, SGLang proposes RadixAttention (Zheng et al., 2024), allowing programmatic control over KV cache reuse. In addition, to address the I/O overhead of reading and writing the KV cache, recent works (Juravsky et al., 2024; Ye et al., 2024; Zhu et al., 2024a; Pan et al., 2025c; Wang et al., 2025i) develop new methods to more efficiently compute both the shared prefix and its multiple completions.

Validation Prompt for AIME

```
You are a rigorous grading instructor for math contest solutions.
You are given the original problem text, the student's full written
    solution (whose final numeric answer is known to be correct),
    and one or more reference solutions (provided only as guidance;
    exact verbatim matching is NOT required).

Your task is to judge whether the student's *reasoning process* is
    logically valid and self-contained, instead of merely checking
    the final answer.

[Original problem]
{ProblemText}

[Student's full solution]
{ModelTrace}

[Reference solution(s) - for guidance only]
Solution 1: ...
Solution 2: ...
Solution 3: ...

Judging criteria:
1) Are key derivations justified with sufficient intermediate steps,
    without circular reasoning or illicit assumptions?
2) Minor arithmetic/notation slips that are explicitly corrected
    later and do not affect the logic may still be acceptable.
3) If there is a fundamental flaw (misused theorem, false equality,
    missing essential conditions) such that the correct final answer
    could be a coincidence, the solution should be judged incorrect
    .
4) Different approaches from the reference are permitted as long as
    the argument forms a logically sound and complete proof.
Please reason step by step before you decide.

Output requirement:
After your analysis, the VERY LAST LINE must be exactly one of the
    following:
[CORRECT]
or
[INCORRECT]
Do not output anything after that final line.
```

Figure 10: Validation prompt used for gpt-5-assisted verification on AIME problems.

---

**Algorithm 1** Efficient Temperature Scaling with Two-Stage Voting

---

**Require:** Questions $\mathcal{Q}$; temperatures $\mathcal{T} = \{T_1, \ldots, T_M\}$; max rounds $R$; intra-temp threshold $\tau_{\text{intra}}$; cross-temp threshold $\tau_{\text{cross}}$; sampler $\text{SAMPLE}(q, T)$; extractor $\text{ANS}(y)$
**Ensure:** For each $q \in \mathcal{Q}$, per-temperature pools $\{\mathcal{S}_{q,T}\}$ and their answers
  1: **for all** $q \in \mathcal{Q}$ **do**
  2:     **for all** $T \in \mathcal{T}$ **do**
  3:         $\mathcal{S}_{q,T} \leftarrow \emptyset$                               $\triangleright$ trace pool at temperature $T$
  4:         $\mathcal{A}_{q,T} \leftarrow \emptyset$                             $\triangleright$ multiset of answers for $T$
  5:     **end for**
  6:     $\text{ACTIVE}(q) \leftarrow$ true
  7: **end for**
  8: **for** $r \leftarrow 1$ **to** $R$ **do**
  9:     **for all** $q \in \mathcal{Q}$ **with** $\text{ACTIVE}(q) =$ true **do**               $\triangleright$ Round-$r$ sampling
10:         **for all** $T \in \mathcal{T}$ **do**
11:             $y \leftarrow \text{SAMPLE}(q, T)$
12:             $a \leftarrow \text{ANS}(y)$
13:             Append $(T, y, a)$ to $\mathcal{S}_{q,T}$; insert $a$ into $\mathcal{A}_{q,T}$
14:         **end for**
                                         $\triangleright$ Stage 1: intra-temperature voting
15:         *all_passed* $\leftarrow$ true;    $\mathcal{V}_q \leftarrow \emptyset$                         $\triangleright$ $\mathcal{V}_q$: one vote per $T$
16:         **for all** $T \in \mathcal{T}$ **do**
17:             Build $h_{q,T}(a) \leftarrow \#\{a' \in \mathcal{A}_{q,T} : a' = a\}$
18:             $v_{\max}^{(T)}(q) \leftarrow \max_a h_{q,T}(a)$
19:             **if** $v_{\max}^{(T)}(q) < \tau_{\text{intra}}$ **then**
20:                 *all_passed* $\leftarrow$ false                       $\triangleright$ this $T$ not confident yet
21:             **else**
22:                 $a_T \leftarrow \arg\max_a h_{q,T}(a)$               $\triangleright$ temperature-$T$ majority
23:                 Append $a_T$ to $\mathcal{V}_q$                    $\triangleright$ one vote from this $T$
24:             **end if**
25:         **end for**
26:         **if** *all_passed* = false **then**
27:             **continue**                 $\triangleright$ skip cross-temp vote; keep sampling next round
28:         **end if**
                                       $\triangleright$ Stage 2: cross-temperature voting
29:         Build cross-temp tally $H_q(b) \leftarrow \#\{a_T \in \mathcal{V}_q : a_T = b\}$
30:         $V_{\max}(q) \leftarrow \max_b H_q(b)$
31:         **if** $V_{\max}(q) \geq \tau_{\text{cross}}$ **then**
32:             $\text{ACTIVE}(q) \leftarrow$ false                           $\triangleright$ early exit for $q$
33:         **end if**
34:     **end for**
35:     **if** $\text{ALLINACTIVE}(\mathcal{Q})$ **then**
36:         **break**
37:     **end if**
38: **end for**
39: **return** $\{\mathcal{S}_{q,T} \mid q \in \mathcal{Q}, T \in \mathcal{T}\}$          $\triangleright$ downstream verifier/BoN picks final answers

---

