# OpenReview forum: "On the Role of Temperature Sampling in Test-Time Scaling"
_ICLR.cc/2026/Conference — Submitted to ICLR 2026_

### Official Review · Reviewer_j3rw · 2025-10-31

**Soundness:** 3
**Presentation:** 3
**Contribution:** 3
**Rating:** 6
**Confidence:** 3

**Summary:**

This paper re-examines the Test-Time Scaling of Large Language Models for reasoning tasks, designed an efficient multi-temperature voting method with early exit for easy questions, reducing computational overhead while preserving performance gains. Experiments across multiple models and datasets validate the effectiveness of the proposed method.

**Strengths:**

1) The comparative experiments are relatively comprehensive; the proposed scheme requires no additional training and is orthogonal to most existing TTS methods.
2) An entropy-based analysis and problem taxonomy are conducted to reveal the dynamic behavior of temperature scaling, constituting a key contribution of this work.

**Weaknesses:**

1) All models used in the comparison belong to the Qwen family, which is insufficient to demonstrate the generality of the proposed method. It is recommended to include models of different families for validation.
2) The main text lacks a formal description of the proposed method, making its exact implementation difficult to grasp.
3) There is no ablation study on the hyper-parameters (e.g., cross-temp threshold vs. intra-temp threshold) and their impact on performance and efficiency, hindering an understanding of how each component contributes to the gains.

**Questions:**

Please refer to the weakness part.

---

> ### Author Response · Authors · 2025-11-27
> **Rebuttal to Reviewer j3rw - Part 1**
>
> We thank Reviewer `j3rw` for the positive assessment and for highlighting both the strengths and the areas that can be improved. Below we respond to each point in detail.
>
> ---
>
> ### [W1] Generality of the Method Beyond Qwen Models
>
> **Reviewer comment:**
> *"All models used in the comparison belong to the Qwen family, which is insufficient to demonstrate the generality of the method. It is recommended to include models of different families for validation."*
>
> ---
>
> **Author response:**
>
> Thank you for pointing this out. We agree that the model scope in the original submission was limited, and that relying only on Qwen3 models does not fully characterize temperature-scaling behavior. In addition, the initial experiments focused primarily on the upper bound rather than equal-budget comparisons. As prior work [1] has emphasized, compute-efficient TTS is essential for a fair and meaningful evaluation. In response, we expand our experimental coverage to include stronger baselines, alternative decoding strategies, and models with RL or complex reasoning capabilities, all evaluated under the same token budget. These additions address the concern and demonstrate that our findings hold across architectures and training paradigms.
>
>
> **Part 1. Comparisons with Other Decoding Methods Under Equivalent Budgets**
>
> To provide a more complete and fair comparison under token budgets, we evaluate seven different decoding configurations:
>
> 1. **Single-temperature (high-temperature T=1.2)**
> 2. **Multi-temperature (uniform grid, T ranging from 0.1 to 1.2)**
> 3. **Single-temperature + Self-Consistency (SC / majority voting)**
> 4. **Multi-temperature + SC**
> 5. **Single-temperature + Verifier**
> 6. **Multi-temperature + Verifier**
> 7. **Multi-temperature + Ours**
>
> Here, SC denotes a self-consistency style early-stop rule that terminates decoding once agreement is reached [2]. Verifier refers to the strongest oracle verifier: as soon as a correct answer is verified, the method stops generating further traces for that question. This isolates model-side effects from verifier quality [3,4] and guarantees fair comparison.
>
> To extend the evaluation beyond the Qwen family, we also include the open-source SOTA models gpt-oss-20B and gpt-oss-120B.
>
> All models are evaluated under a fixed compute budget of 0.1M generated tokens per question. The resulting performance comparison is summarized in the following table.
>
> **Table 1: Performance of various decoding strategies on AIME 2025, evaluated under the same compute budget.**
> | **Model**               | **Single-temp** | **Multi-temp** | **Single-temp + SC** | **Multi-temp + SC** | **Multi-temp + Ours** | **Single-temp + Verifier** | **Multi-temp + Verifier** |
> |-------------------------|------------------|------------------|------------------------|------------------------|--------------------------|------------------------------|-----------------------------|
> | **Qwen3-4B**            | 43.41           | 45.94           | 43.41                 | 45.94                 | **47.49**                     | 45.85                       | 49.34                      |
> | **Qwen3-8B**            | 39.81           | 43.17           | 39.84                 | 43.21                 | **45.00**                     | 41.50                       | 45.40                      |
> | **gpt-oss-20B**         | 83.27           | 87.17           | 83.27                 | 87.17                 | **87.17**                     | 84.59                       | 90.59                      |
> | **gpt-oss-120B**        | 92.92           | 93.15           | 92.94                 | 93.17                 | **93.23**                     | 93.33                       | 93.58                      |
>
> **Multi-temperature sampling helps.** On AIME 2025, even the plain multi-temperature strategy (simply splitting the same budget across several temperatures) outperforms the single-temperature setting. This means different temperatures indeed solve different questions, and by scaling across temperatures we lose nothing while gaining better performance under the same budget.
>
> **Our early exit method is stronger than SC, and multi-temperature sampling remains better when we have a verifier.** When no verifier is available, our method outperforms SC. When a strong verifier is available, decoding stops immediately once a correct answer is verified, saving substantial compute. Multi-temperature sampling still achieves higher accuracy. The gap between single-temp and multi-temp varies: sometimes small, sometimes very large. This is because we cannot reliably know whether the chosen single temperature is the "good" one for a given model or dataset, which is why temperature scaling is needed.

---

> > ### Author Response · Authors · 2025-11-27
> > **Rebuttal to Reviewer j3rw - Part 2**
> >
> > **Part 2. Temperature Scaling for RL and Thinking Models**
> >
> > Here we incorporate two models: Polaris-4B (RL-trained) and Qwen3-4B-Thinking (reasoning mode). These models naturally produce longer and more structured reasoning traces, making them a good testbed for whether temperature scaling continues to help under stronger reasoning capabilities. The results are shown in Table 2.
> >
> > **Table 2: AIME 2025 results for Polaris-4B and Qwen3-4B-Thinking.**
> > | **Model**               | **Single-temp** | **Multi-temp** | **Single-temp + SC** | **Multi-temp + SC** | **Multi-temp + Ours** | **Single-temp + Verifier** | **Multi-temp + Verifier** |
> > |-------------------------|------------------|------------------|------------------------|------------------------|--------------------------|------------------------------|-----------------------------|
> > | **Polaris-4B**          | 86.67             | 89.80             | 86.67                   | 90.67                   | **91.13**|86.67                     | 96.67                                                 |
> > | **Qwen3-4B-Thinking**   | 86.67             | 87.65             | 86.67                   | 87.88                   | **88.33**                     | 86.67                         | 90.00                        |
> >
> > **Temperature scaling also improves RL and thinking models.** Under the same compute budget, multi-temperature sampling outperforms single-temperature sampling, and the gap becomes large under a strong verifier (e.g., Polaris-4B improves by 10 points). This indicates that single-temperature sampling fails to explore many valid reasoning paths, while temperature scaling recovers that missing capability under same budget.
> >
> > **RL is strong, and temperature scaling makes it even stronger.**  Prior work [5] suggested that RL-trained models may not provide fundamentally new capabilities, since a base model with extensive test-time sampling can match or surpass them. Our results give a more nuanced picture: Polaris-4B gains additional accuracy when combined with temperature scaling, indicating that RL *does* introduce new reasoning modes that a single temperature cannot reliably activate. (This is expected, since the verifier reduces uncertainty.) At the same time, a base model equipped with temperature scaling can exceed the single-temperature RL model, suggesting complementary strengths. We view RL and temperature scaling as compatible rather than competing, frameworks such as TTRL [6] and Reasoning with Sampling [7] can combine both to further enhance reasoning performance.
> >
> > ---
> >
> > ### [W2] Lack of Formal Description of the Proposed Method in the Main Text
> >
> > **Reviewer comment:**
> > *"The main text lacks a formal description of the proposed method, making its exact implementation difficult to grasp."*
> >
> > ---
> >
> > **Author response:**
> >
> > Thank you for pointing out that the methodological details were not sufficiently described in the original submission. Due to space constraints, our explanations for temperature selection and early-exit design were brief, which may have caused confusion. In the revised version, we will incorporate following contents.
> >
> > **Part 1. Principled Selection of A Temperature Support Set**
> > Given a grid of candidate temperatures
> > \begin{align*}
> > \mathcal{T}_ {\text{grid}} = \lbrace 0.0, 0.1, \ldots, 1.2 \rbrace
> > \end{align*}
> > and a fixed budget $K$, we first compute for each temperature $T \in \mathcal{T}_ {\text{grid}}$ its solvable set
> > \begin{align*}
> > \mathcal{S}(T) = \lbrace q : \mathrm{Pass@K}(q;T) \ge \epsilon \rbrace,
> > \end{align*}
> > using the same definition as in [W1Q1]. Let
> > \begin{align*}
> > \mathcal{U} = \bigcup_{T \in \mathcal{T}_ {\text{grid}}} \mathcal{S}(T)
> > \end{align*}
> > be the full union over all temperatures. Our goal is to find a small *support set* of temperatures $\mathcal{T}_ {\text{sup}} \subseteq \mathcal{T}_ {\text{grid}}$ such that
> > \begin{align*}
> > \Big|\bigcup_{T \in \mathcal{T}_ {\text{sup}}} \mathcal{S}(T)\Big|
> > \ge \rho  |\mathcal{U}|,
> > \end{align*}
> > for a target coverage $\rho$ (e.g., $\rho = 0.99$). This is exactly a set cover problem, which we solve with the standard greedy algorithm: starting from $\mathcal{T}_ {\text{sup}} = \emptyset$, at each step we add the temperature
> > \begin{align*}
> > T^* = \arg\max_{T \in \mathcal{T}_ {\text{grid}} \setminus \mathcal{T}_ {\text{sup}}}
> > \Big|\mathcal{S}(T) \setminus \bigcup_{T' \in \mathcal{T}_ {\text{sup}}} \mathcal{S}(T')\Big|,
> > \end{align*}
> > and stop when the coverage constraint above is met. Applying this procedure across our models and datasets, we obtain a support set $[0.4, 1.2]$ that recovers almost all solvable problems. This is the temperature set used in our multi-temperature experiments.

---

> ### Author Response · Authors · 2025-11-27
> **Rebuttal to Reviewer j3rw - Part 3**
>
> **Part 2. Early-Exit Mechanism**
> For a given temperature $T$ and a fixed per-temperature budget $K_T$, decoding produces a sequence of candidate answers $\lbrace a_1, a_2, \dots \rbrace$. Our early-exit rule monitors the vote distribution
> $$
> p_t(y) = \frac{1}{t}\sum_{i=1}^t \mathbf{1}[a_i = y],
> $$
> and terminates decoding at the first step $t < K_T$ where either of the following conditions holds:
>
> (1) **Intra-temperature agreement:**
>    $$\max_y p_t(y) \ge \tau_{\text{intra}},$$
>    indicating that the samples at temperature $T$ have converged.
>
> (2) **Cross-temperature agreement:** if the current temperature $T$ is not the first one in the schedule, we additionally maintain the running majority vote across all temperatures processed so far. Let
> \begin{align*}
> p_t^{\le T}(y) = \text{normalized vote count across all visited temperatures},
> \end{align*}
> then we exit early if
> \begin{align*}
> \max_y p_{t}^{\le T}(y) \ge  \tau_{\text{cross}}.
> \end{align*}
>
> If neither condition is triggered, decoding continues until either $K_T$ samples are consumed or a verifier (when available) confirms correctness and forces immediate termination. The thresholds $\tau_{\text{intra}}$ and $\tau_{\text{cross}}$ are kept constant across all models and datasets; in the following [W4] part, we will include a sensitivity analysis showing the method is robust across a wide range of values.
>
> This mechanism generalizes SC by enabling consensus checks not only *within* a single temperature, but also *across multiple temperatures*. It ensures compute is concentrated on harder questions while retaining the benefits of temperature scaling.
>
> ---
>
> ### [W3] Missing Ablation Study on Hyperparameters
>
> **Reviewer comment:**
> *"There is no ablation study on the hyper-parameters (e.g., cross-temp threshold vs. intra-temp threshold) and their impact on performance and efficiency."*
>
> ---
>
> **Author response:**
>
> Thank you for the suggestion. We sweep the intra-temperature threshold $\tau_{\text{intra}}$ with $\tau_{\text{cross}} = 1.0$. Accuracy (in %) is shown below:
>
> **Table 3: Sweeping $\tau_{\text{intra}}$, Qwen3-4B on AIME 2025.**
> | $\tau_{\text{intra}}$ | 0.0  | 0.1  | 0.2  | 0.3  | 0.4  | 0.5  | 0.6  | 0.7  | 0.8  | 0.9  | 1.0  |
> |------------------------|------|------|------|------|------|------|------|------|------|------|------|
> | Accuracy (%)           |45.50 |45.50 |46.94 |47.26 |47.17 |47.03 |47.62 |47.57 |47.57 |47.35 |47.31 |
>
> We also fix $\tau_{\text{intra}} = 0.8$ and sweep $\tau_{\text{cross}}$, and the resulting accuracy remains unchanged (47.57% for all tested values).
>
> **These results show that the early-exit mechanism is stable:** varying either threshold across a wide range barely changes accuracy. This confirms the method is not sensitive to hyperparameters and simply triggers once the vote distribution becomes confidently peaked.

---

> > ### Author Response · Authors · 2025-11-27
> > **Rebuttal to Reviewer j3rw - Part 4**
> >
> > ### Connection to Previous Temperature-Sampling Literature
> >
> > Existing work on dynamic temperature sampling is largely **optimization-based**: the goal is to infer a single "best" temperature for the entire dataset or for each question. For example, prior work [8] estimates the optimal temperature from dataset-level entropy, which still results in one global temperature; as we show, restricting to a single temperature can severely limit performance on hard problems. Other approaches [9] train an adapter to predict an optimal temperature, yet in the evaluation such methods predict the same temperature (0.0) for MATH500, and clearly, this is suboptimal. In short, these methods attempt to **optimize a single temperature**, while our results indicate that such a notion of "best" temperature may not exist.
> >
> > Our findings suggest that different problems favor different temperatures because temperature reshapes the autoregressive trajectory distribution in problem-specific ways. Instead of optimizing for a single value, we therefore adopt a **scaling-based** approach: using a set of temperatures that jointly cover the union of solvable problems. With a verifier, easy problems exit immediately at the first temperature that produces a correct trace, which drastically reduces the effective cost of temperature scaling. As shown in Tables 1 and 3, under an equal token budget, multi-temperature sampling consistently outperforms single-temperature approaches.
> >
> > ---
> >
> > ### References
> >
> > 1. Liu et al., Can 1B LLM Surpass 405B LLM? Rethinking Compute-Optimal Test-Time Scaling, Reasoning and Planning for LLMs @ ICLR 2025
> > 2. Xue et al., Dynamic Voting for Efficient Reasoning in Large Language Models, EMNLP Findings 2023.
> > 3. Zhang et al., OpenPRM: Building Open-domain Process-based Reward Models with Preference Trees, ICLR 2025
> > 4. Lee et al., Rethinking Reward Models for Multi-Domain Test-Time Scaling, Arxiv 2025
> > 5. Yue et al., Does Reinforcement Learning Really Incentivize Reasoning Capacity in LLMs Beyond the Base Model, NeurIPS 2025
> > 6. Zuo et al., TTRL: Test-Time Reinforcement Learning, NeurIPS 2025
> > 7. Karan et al., Reasoning with Sampling: Your Base Model is Smarter Than You Think, Arxiv 2025
> > 8. Du et al., Optimizing Temperature for Language Models with Multi-Sample Inference, ICML 2025
> > 9. Dhuliawala et al., Adaptive Decoding via Latent Preference Optimization, Arxiv 2024
> >
> >
> >
> > ---
> >
> > We appreciate Reviewer `j3rw` for the encouraging comments and constructive suggestions. We hope the additional clarifications and revisions will fully resolve the concerns raised.

---

### Official Review · Reviewer_3DQD · 2025-11-01

**Soundness:** 2
**Presentation:** 3
**Contribution:** 2
**Rating:** 4
**Confidence:** 4

**Summary:**

This paper investigates the role of temperature sampling in the test-time scaling (TTS) paradigm for large language models (LLMs). Traditionally, TTS boosts reasoning performance by generating multiple samples and selecting the best one via a verifier. The authors show that beyond a certain K, accuracy plateaus, and some problems remain unsolved regardless of further sampling. The key insight is that different sampling temperatures T lead to the solution of different subsets of problems, implying that a single-temperature TTS explores only a part of the model’s reasoning space. The authors propose multi-temperature scaling, where samples are drawn from multiple temperatures to expand the model’s “reasoning boundary.” They show empirically that temperature scaling allows base models to achieve performance comparable to RL-trained models, without additional fine-tuning.

**Strengths:**

- The observation is interesting and novel.
- The figures are informative and well-presented.
- The authors conduct experiments across multiple domains.
- The paper is easy to follow.

**Weaknesses:**

- The major concern is that the experiments are only conducted on Qwen3 series models. However, different models may have different properties regarding temperature scaling. For example, the recommended temperature for Qwen3 and DeepSeek-R1-Distill series models are different [1].
- The experiments are only restricted to models up to 8B parameters. However, larger models (e.g., 70B+) may have different behaviors with respect to temperature scaling.
- The paper lacks a theoretical framework explaining why certain temperatures preferentially solve specific hard problems.

[1] POLARIS: A POst-training recipe for scaling reinforcement Learning on Advanced ReasonIng modelS.

**Questions:**

- Can the authors give some high-level points of why using different temperatures helps? For example, is it because different temperatures lead to more diverse reasoning paths, or because certain temperatures are better suited for specific types of problems?

---

> ### Author Response · Authors · 2025-11-27
> **Rebuttal to Reviewer 3DQD - Part 1**
>
> We sincerely thank Reviewer `3DQD` for the clear summary and constructive feedback. Below we address each concern point-by-point.
>
> ---
>
> ### [W1] Experiments Limited to the Qwen3 Series
>
> **Reviewer comment:**
> *"The major concern is that the experiments are only conducted on Qwen3 series models. Different models may have different properties regarding temperature scaling. For example, the recommended temperature for Qwen3 and DeepSeek-R1-Distill series models are different."*
>
> ---
>
> **Author response:**
>
> Thank you for pointing this out. We agree that the model scope in the original submission was limited, and that relying only on Qwen3 models does not fully characterize temperature-scaling behavior. In addition, the initial experiments focused primarily on the upper bound rather than equal-budget comparisons. As prior work [1] has emphasized, compute-efficient TTS is essential for a fair and meaningful evaluation. In response, we expand our experimental coverage to include stronger baselines, alternative decoding strategies, and models with RL or complex reasoning capabilities, all evaluated under the same token budget. These additions address the concern and demonstrate that our findings hold across architectures and training paradigms.
>
>
> **Part 1. Comparisons with Other Decoding Methods Under Equivalent Budgets**
>
> To provide a more complete and fair comparison under token budgets, we evaluate seven different decoding configurations:
>
> 1. **Single-temperature (high-temperature T=1.2)**
> 2. **Multi-temperature (uniform grid, T ranging from 0.1 to 1.2)**
> 3. **Single-temperature + Self-Consistency (SC / majority voting)**
> 4. **Multi-temperature + SC**
> 5. **Single-temperature + Verifier**
> 6. **Multi-temperature + Verifier**
> 7. **Multi-temperature + Ours**
>
> Here, SC denotes a self-consistency style early-stop rule that terminates decoding once agreement is reached [2]. Verifier refers to the strongest oracle verifier: as soon as a correct answer is verified, the method stops generating further traces for that question. This isolates model-side effects from verifier quality [3,4] and guarantees fair comparison.
>
> To extend the evaluation beyond the Qwen family, we also include the open-source SOTA models gpt-oss-20B and gpt-oss-120B.
>
> All models are evaluated under a fixed compute budget of 0.1M generated tokens per question. The resulting performance comparison is summarized in the following table.
>
> **Table 1: Performance of various decoding strategies on AIME 2025, evaluated under the same compute budget.**
> | **Model**               | **Single-temp** | **Multi-temp** | **Single-temp + SC** | **Multi-temp + SC** | **Multi-temp + Ours** | **Single-temp + Verifier** | **Multi-temp + Verifier** |
> |-------------------------|------------------|------------------|------------------------|------------------------|--------------------------|------------------------------|-----------------------------|
> | **Qwen3-4B**            | 43.41           | 45.94           | 43.41                 | 45.94                 | **47.49**                     | 45.85                       | 49.34                      |
> | **Qwen3-8B**            | 39.81           | 43.17           | 39.84                 | 43.21                 | **45.00**                     | 41.50                       | 45.40                      |
> | **gpt-oss-20B**         | 83.27           | 87.17           | 83.27                 | 87.17                 | **87.17**                     | 84.59                       | 90.59                      |
> | **gpt-oss-120B**        | 92.92           | 93.15           | 92.94                 | 93.17                 | **93.23**                     | 93.33                       | 93.58                      |
>
> **Multi-temperature sampling helps.** On AIME 2025, even the plain multi-temperature strategy (simply splitting the same budget across several temperatures) outperforms the single-temperature setting. This means different temperatures indeed solve different questions, and by scaling across temperatures we lose nothing while gaining better performance under the same budget.
>
> **Our early exit method is stronger than SC, and multi-temperature sampling remains better when we have a verifier.** When no verifier is available, our method outperforms SC. When a strong verifier is available, decoding stops immediately once a correct answer is verified, saving substantial compute. Multi-temperature sampling still achieves higher accuracy. The gap between single-temp and multi-temp varies: sometimes small, sometimes very large. This is because we cannot reliably know whether the chosen single temperature is the "good" one for a given model or dataset, which is why temperature scaling is needed.

---

> > ### Author Response · Authors · 2025-11-27
> > **Rebuttal to Reviewer 3DQD - Part 2**
> >
> > **Part 2. Temperature Scaling for RL and Thinking Models**
> >
> > Here we incorporate two models: Polaris-4B (RL-trained) and Qwen3-4B-Thinking (reasoning mode). These models naturally produce longer and more structured reasoning traces, making them a good testbed for whether temperature scaling continues to help under stronger reasoning capabilities. The results are shown in Table 2.
> >
> > **Table 2: AIME 2025 results for Polaris-4B and Qwen3-4B-Thinking.**
> > | **Model**               | **Single-temp** | **Multi-temp** | **Single-temp + SC** | **Multi-temp + SC** | **Multi-temp + Ours** | **Single-temp + Verifier** | **Multi-temp + Verifier** |
> > |-------------------------|------------------|------------------|------------------------|------------------------|--------------------------|------------------------------|-----------------------------|
> > | **Polaris-4B**          | 86.67             | 89.80             | 86.67                   | 90.67                   | **91.13**|86.67                     | 96.67                                                 |
> > | **Qwen3-4B-Thinking**   | 86.67             | 87.65             | 86.67                   | 87.88                   | **88.33**                     | 86.67                         | 90.00                        |
> >
> > **Temperature scaling also improves RL and thinking models.** Under the same compute budget, multi-temperature sampling outperforms single-temperature sampling, and the gap becomes large under a strong verifier (e.g., Polaris-4B improves by 10 points). This indicates that single-temperature sampling fails to explore many valid reasoning paths, while temperature scaling recovers that missing capability under same budget.
> >
> > **RL is strong, and temperature scaling makes it even stronger.**  Prior work [5] suggested that RL-trained models may not provide fundamentally new capabilities, since a base model with extensive test-time sampling can match or surpass them. Our results give a more nuanced picture: Polaris-4B gains additional accuracy when combined with temperature scaling, indicating that RL *does* introduce new reasoning modes that a single temperature cannot reliably activate. (This is expected, since the verifier reduces uncertainty.) At the same time, a base model equipped with temperature scaling can exceed the single-temperature RL model, suggesting complementary strengths. We view RL and temperature scaling as compatible rather than competing, frameworks such as TTRL [6] and Reasoning with Sampling [7] can combine both to further enhance reasoning performance.
> >
> >
> > ---
> >
> > ### [W2] Experiments Limited to Models up to 8B Parameters
> >
> > **Reviewer comment:**
> > *"The experiments are restricted to models up to 8B parameters. Larger models (e.g., 70B+) may behave differently under temperature scaling."*
> >
> > ---
> >
> > **Author response:**
> >
> > Thank you for raising this concern. We have expanded our experiments to include larger open-source models (gpt-oss-20B and gpt-oss-120B), as reported in [W1]. Interestingly, temperature scaling becomes even more effective for larger models: most easy problems exit early at the first temperature that produces a correct trace, allowing the remaining budget to be concentrated on harder problems. Under the same token budget, gpt-oss-20B improves by more than 6 points with multi-temperature sampling, and gpt-oss-120B also shows consistent gains. These results indicate that temperature scaling remains beneficial, and often more suitable, for larger models as well.
> >
> >
> > ---
> >
> > ### [W3] Lack of Theoretical Framework Explaining Temperature–Problem Dependency
> >
> > **Reviewer comment:**
> > *"The paper lacks a theoretical framework explaining why certain temperatures preferentially solve specific hard problems."*
> >
> > ---
> >
> > **Author response:**
> >
> > **Why different temperatures solve different problems.**  Formally, an LLM defines a distribution over full reasoning trajectories
> > $$
> > p_T(y_{1:T} \mid q)
> > = \prod_{t=1}^T p_T(y_t \mid y_{<t}, q),
> > \qquad
> > p_T(\cdot \mid y_{<t},q)=\mathrm{softmax}\!\left(\frac{z_t(y_{<t},q)}{T}\right),
> > $$
> > where changing the temperature $T$ rescales every token-level logit at every timestep. Even small changes in the token distribution at a few steps can compound through the autoregressive process and substantially alter the entire trajectory.
> >
> > For a given question $q$, let $\mathcal{Y}^* (q)$ denote the set of trajectories that lead to a correct answer. Under a finite sample budget $K$, the probability of solving $q$ at temperature $T$ is
> > \begin{align*}
> > \Pr[\text{solve } q \mid T,K]
> > = 1 - \big(1 - p_T(\mathcal{Y}^* (q))\big)^K,
> > \qquad
> > p_T(\mathcal{Y}^* (q))
> > =\sum_{y\in\mathcal{Y}^* (q)} p_T(y\mid q).
> > \end{align*}

---

> ### Author Response · Authors · 2025-11-27
> **Rebuttal to Reviewer 3DQD - Part 3**
>
> Because $p_T(y\mid q)$ depends multiplicatively on the rescaled softmax at each step,
> \begin{align*}
> \frac{p_T(i \mid y_{<t},q)}{p_T(j \mid y_{<t},q)}
> = \exp\left(\frac{z_i - z_j}{T}\right),
> \end{align*}
> changing $T$ perturbs the relative probability of taking "correct-leading" vs. "incorrect-leading" continuations at the key branching steps of the reasoning process. Since each question has its own logit landscape and its own set of branching points, the temperature $T$ that maximizes $p_T(\mathcal{Y}^*(q))$, and thus the success probability under finite $K$, varies across problems.
>
> In short, the temperature-problem dependency arises because temperature reshapes the autoregressive probability of entire reasoning trajectories, and each question's correct trajectories occupy different regions of this landscape. Multi-temperature sampling recovers these different regions in a compute-controlled way.
>
>
> ---
>
>
>
> ### [Q1] High-Level Explanation of Why Different Temperatures Help
>
> **Reviewer question:**
> *"Can the authors give high-level points of why using different temperatures helps? For example, is it diversity of reasoning paths, or suitability for specific problem types?"*
>
> ---
>
> **Author response:**
>
> Yes, the key intuition is that temperature reshapes the entire autoregressive reasoning trajectory, and each question has different "branching points" where the model must choose among several plausible continuations. Different temperatures emphasize different parts of this branching landscape, leading to qualitatively different reasoning paths.
>
> Existing dynamic-temperature approaches are mostly **optimization-based**: they attempt to infer a *single* "best’’ temperature for each dataset or each question (e.g., by entropy heuristics [8] or by using an adapter to predict an optimal value [9]). However, as we show in our experiments, such a universal best temperature often does not exist, hard problems typically require different temperatures from easy ones, and the adapter-based methods frequently collapse to predicting a single extreme value (e.g., 0.0 on MATH500), which is clearly suboptimal.
>
> Our approach is instead **scaling-based**: rather than committing to one temperature, we use a set of temperatures that jointly cover the union of solvable reasoning trajectories. With a verifier, easy questions exit as soon as one temperature yields a correct trace, making the effective budget comparable to single-temperature sampling. Under equal compute, this multi-temperature scaling consistently outperforms optimization-based single-temperature methods.
>
>
> ---
>
> ### References
>
> 1. Liu et al., Can 1B LLM Surpass 405B LLM? Rethinking Compute-Optimal Test-Time Scaling, Reasoning and Planning for LLMs @ ICLR 2025
> 2. Xue et al., Dynamic Voting for Efficient Reasoning in Large Language Models, EMNLP Findings 2023.
> 3. Zhang et al., OpenPRM: Building Open-domain Process-based Reward Models with Preference Trees, ICLR 2025
> 4. Lee et al., Rethinking Reward Models for Multi-Domain Test-Time Scaling, Arxiv 2025
> 5. Yue et al., Does Reinforcement Learning Really Incentivize Reasoning Capacity in LLMs Beyond the Base Model, NeurIPS 2025
> 6. Zuo et al., TTRL: Test-Time Reinforcement Learning, NeurIPS 2025
> 7. Karan et al., Reasoning with Sampling: Your Base Model is Smarter Than You Think, Arxiv 2025
> 8. Du et al., Optimizing Temperature for Language Models with Multi-Sample Inference, ICML 2025
> 9. Dhuliawala et al., Adaptive Decoding via Latent Preference Optimization, Arxiv 2024
>
>
> ---
>
>
>
> We thank Reviewer `3DQD` again for the insightful and helpful comments. We hope our clarifications and revisions will adequately address the concerns raised.

---

### Official Review · Reviewer_zrJ8 · 2025-11-02

**Soundness:** 3
**Presentation:** 2
**Contribution:** 2
**Rating:** 4
**Confidence:** 3

**Summary:**

This paper explores the limitations of current test-time scaling (TTS) methods, which typically sample at only a single temperature. The authors propose scaling along the temperature dimension—that is, sampling at various different temperatures—to capture the union of all problems the model is capable of solving. The experiments demonstrate that this simple "temperature scaling" approach can allow a base model to match or even exceed the performance of models fine-tuned with RL.

**Strengths:**

- The paper's core hypothesis and observations are concise and meaningful. The idea that a model's set of solvable problems is temperature-dependent, and that the union of these sets represents the model's true capability, is a valuable insight.

- The figures in the paper are clear, well-executed, and effectively support the analysis and conclusions. Figure 3, in particular, does an excellent job of visualizing and explaining the core findings.

**Weaknesses:**

- High Computational Cost: The method is computationally expensive as it requires extensive sampling across a wide range of different temperatures to achieve its full effect.

- Lack of Deep Explanation: The paper lacks a deep explanation for why different problems require different temperatures to be solved. The analysis describes the resulting entropy dynamics (what is happening) but doesn't fully explain the root cause (why this dependency exists). Entropy seems to be a consequence of this phenomenon, not the cause.

- Novelty of Claims: The conclusion that a base model with sufficient repeated sampling can outperform an RL-tuned model is not an entirely novel finding in the field.

- Potential for Cherry-picking in Figure 3d: It is unclear how the specific value of $k$ (e.g., $k=128$) was chosen for the Pass@k comparison in Figure 3d. It is plausible that as $k$ increases, repeated sampling at any single temperature might eventually surpass the RL model's performance. The comparison at $k=128$ feels somewhat selective and may not present the full picture.

**Questions:**

There is a body of existing work focused on teaching models to perform dynamic temperature sampling (i.e., learning to adjust temperature during generation). Could the authors discuss this line of research and elaborate on its connection to their findings?

---

> ### Author Response · Authors · 2025-11-27
> **Rebuttal to Reviewer zrJ8 - Part 1**
>
> We deeply appreciate Reviewer `zrJ8`’s thoughtful and constructive feedback. Below, we address each concern point-by-point.
>
> ---
>
> ### [W1] High Computational Cost of Multi-Temperature Sampling
>
> **Reviewer comment:**
> *"The method is computationally expensive as it requires extensive sampling across a wide range of different temperatures to achieve its full effect."*
>
> ---
>
> **Author response:**
>
> Thank you for raising the concern about computational cost. The original submission focused on establishing the upper bound of test-time scaling, but prior work [1] has shown that compute-efficient TTS is equally important for a fair and meaningful comparison. In response, we now include extensive equal-budget evaluations across all settings.  These additional experiments directly address the cost concern and demonstrate that multi-temperature sampling continues to outperform single-temperature sampling even when computation is held constant.
>
>
> We evaluate seven different decoding configurations:
>
> 1. **Single-temperature (high-temperature T=1.2)**
> 2. **Multi-temperature (uniform grid, T ranging from 0.1 to 1.2)**
> 3. **Single-temperature + Self-Consistency (SC / majority voting)**
> 4. **Multi-temperature + SC**
> 5. **Single-temperature + Verifier**
> 6. **Multi-temperature + Verifier**
> 7. **Multi-temperature + Ours**
>
> Here, SC denotes a self-consistency style early-stop rule that terminates decoding once agreement is reached [2]. Verifier refers to the strongest oracle verifier: as soon as a correct answer is verified, the method stops generating further traces for that question. This isolates model-side effects from verifier quality [3,4] and guarantees fair comparison.
>
> To extend the evaluation beyond the Qwen family, we also include the open-source SOTA models gpt-oss-20B and gpt-oss-120B.
>
> All models are evaluated under a fixed compute budget of 0.1M generated tokens per question. The resulting performance comparison is summarized in the following table.
>
> **Table 1: Performance of various decoding strategies on AIME 2025, evaluated under the same compute budget.**
> | **Model**               | **Single-temp** | **Multi-temp** | **Single-temp + SC** | **Multi-temp + SC** | **Multi-temp + Ours** | **Single-temp + Verifier** | **Multi-temp + Verifier** |
> |-------------------------|------------------|------------------|------------------------|------------------------|--------------------------|------------------------------|-----------------------------|
> | **Qwen3-4B**            | 43.41           | 45.94           | 43.41                 | 45.94                 | **47.49**                     | 45.85                       | 49.34                      |
> | **Qwen3-8B**            | 39.81           | 43.17           | 39.84                 | 43.21                 | **45.00**                     | 41.50                       | 45.40                      |
> | **gpt-oss-20B**         | 83.27           | 87.17           | 83.27                 | 87.17                 | **87.17**                     | 84.59                       | 90.59                      |
> | **gpt-oss-120B**        | 92.92           | 93.15           | 92.94                 | 93.17                 | **93.23**                     | 93.33                       | 93.58                      |
>
> **Multi-temperature sampling helps.** On AIME 2025, even the plain multi-temperature strategy (simply splitting the same budget across several temperatures) outperforms the single-temperature setting. This means different temperatures indeed solve different questions, and by scaling across temperatures we lose nothing while gaining better performance under the same budget.
>
> **Our early exit method is stronger than SC, and multi-temperature sampling remains better when we have a verifier.** When no verifier is available, our method outperforms SC. When a strong verifier is available, decoding stops immediately once a correct answer is verified, saving substantial compute. Multi-temperature sampling still achieves higher accuracy. The gap between single-temp and multi-temp varies: sometimes small, sometimes very large. This is because we cannot reliably know whether the chosen single temperature is the "good" one for a given model or dataset, which is why temperature scaling is needed.

---

> > ### Author Response · Authors · 2025-11-27
> > **Rebuttal to Reviewer zrJ8 - Part 2**
> >
> > ### [W2] Lack of Deeper Explanation for Temperature–Problem Dependency
> >
> > **Reviewer comment:**
> > *"The paper lacks a deep explanation for why different problems require different temperatures to be solved. The analysis describes what is happening (entropy) but not why the dependency exists. Entropy seems to be a consequence, not the cause."*
> >
> > ---
> >
> > **Author response:**
> >
> > **Our entropy analysis was not intended as an explanation for the temperature–problem dependency.**  Thank you for pointing this out, and you are correct that entropy describes *what* happens rather than *why*. In the paper, the entropy curves were used only to show that for easy questions, entropy loosely correlates with correctness, while for hard questions this correlation breaks down. We do not intend to claim that entropy is the cause of the temperature-problem dependency, and we will make this explicit in the revised version.
> >
> > **Why different temperatures solve different problems.**  Formally, an LLM defines a distribution over full reasoning trajectories
> > $$
> > p_T(y_{1:T} \mid q)
> > = \prod_{t=1}^T p_T(y_t \mid y_{<t}, q),
> > \qquad
> > p_T(\cdot \mid y_{<t},q)=\mathrm{softmax}\!\left(\frac{z_t(y_{<t},q)}{T}\right),
> > $$
> > where changing the temperature $T$ rescales every token-level logit at every timestep. Even small changes in the token distribution at a few steps can compound through the autoregressive process and substantially alter the entire trajectory.
> >
> > For a given question $q$, let $\mathcal{Y}^* (q)$ denote the set of trajectories that lead to a correct answer. Under a finite sample budget $K$, the probability of solving $q$ at temperature $T$ is
> > \begin{align*}
> > \Pr[\text{solve } q \mid T,K]
> > = 1 - \big(1 - p_T(\mathcal{Y}^* (q))\big)^K,
> > \qquad
> > p_T(\mathcal{Y}^* (q))
> > =\sum_{y\in\mathcal{Y}^* (q)} p_T(y\mid q).
> > \end{align*}
> >
> > Because $p_T(y\mid q)$ depends multiplicatively on the rescaled softmax at each step,
> > \begin{align*}
> > \frac{p_T(i \mid y_{<t},q)}{p_T(j \mid y_{<t},q)}
> > = \exp\left(\frac{z_i - z_j}{T}\right),
> > \end{align*}
> > changing $T$ perturbs the relative probability of taking "correct-leading" vs. "incorrect-leading" continuations at the key branching steps of the reasoning process. Since each question has its own logit landscape and its own set of branching points, the temperature $T$ that maximizes $p_T(\mathcal{Y}^*(q))$, and thus the success probability under finite $K$, varies across problems.
> >
> > In short, the temperature-problem dependency arises because temperature reshapes the autoregressive probability of entire reasoning trajectories, and each question's correct trajectories occupy different regions of this landscape. Multi-temperature sampling recovers these different regions in a compute-controlled way.
> >
> >
> >
> > ---
> >
> > ### [W3] Novelty of Claims Regarding Base Models Matching RL-Tuned Models
> >
> > **Reviewer comment:**
> > *"The conclusion that a base model with sufficient repeated sampling can outperform an RL-tuned model is not entirely novel."*
> >
> > ---
> >
> > **Author response:**
> >
> > Thank you for raising this point. Prior work [5] has suggested that a base model can match an RL-tuned model under a single-temperature TTS setting, but we find this conclusion does not generally hold. A key confound is that TTS may occasionally hit the right answer while following an incorrect or logically invalid reasoning path. To quantify this, we use GPT-5 and reference answers as an external verifier to evaluate trace correctness for all problems with accuracy below 3%. The results show that base-model traces are far from reliably correct:
> >
> > **Table 2: Trace correctness (%) on low-accuracy problems**
> >
> > | Model             | Qwen3-0.6B | Qwen3-1.7B | Qwen3-4B | Qwen3-8B | gpt-oss-20B | gpt-oss-120B |
> > |-------------------|------------|------------|----------|----------|-------------|---------------|
> > | Correctness (%)   |   24.14    |   47.11    |  30.00   |  26.62   |    20.92    |     45.56     |
> >
> > These results show that the base model's *actual* reasoning correctness is much lower than suggested by raw Pass@1 accuracy, and in this sense base models do **not** match RL-tuned models under single-temperature TTS. Our claim is that RL models remain strong, but temperature scaling combined with verification enables base models to *match the single-temperature RL model*, whereas single-temperature base models cannot, as shown in Fig. 3d. In the following, we incorporate RL models into the comparison to make this distinction explicit.

---

> > > ### Author Response · Authors · 2025-11-27
> > > **Rebuttal to Reviewer zrJ8 - Part 3**
> > >
> > > Here we incorporate two models: Polaris-4B (RL-trained) and Qwen3-4B-Thinking (reasoning mode). These models naturally produce longer and more structured reasoning traces, making them a good testbed for whether temperature scaling continues to help under stronger reasoning capabilities. The results are shown in Table 3.
> > >
> > > **Table 3: AIME 2025 results for Polaris-4B and Qwen3-4B-Thinking.**
> > > | **Model**               | **Single-temp** | **Multi-temp** | **Single-temp + SC** | **Multi-temp + SC** | **Multi-temp + Ours** | **Single-temp + Verifier** | **Multi-temp + Verifier** |
> > > |-------------------------|------------------|------------------|------------------------|------------------------|--------------------------|------------------------------|-----------------------------|
> > > | **Polaris-4B**          | 86.67             | 89.80             | 86.67                   | 90.67                   | **91.13**|86.67                     | 96.67                                                 |
> > > | **Qwen3-4B-Thinking**   | 86.67             | 87.65             | 86.67                   | 87.88                   | **88.33**                     | 86.67                         | 90.00                        |
> > >
> > > **Temperature scaling also improves RL and thinking models.** Under the same compute budget, multi-temperature sampling outperforms single-temperature sampling, and the gap becomes large under a strong verifier (e.g., Polaris-4B improves by 10 points). This indicates that single-temperature sampling fails to explore many valid reasoning paths, while temperature scaling recovers that missing capability under same budget.
> > >
> > > **RL is strong, and temperature scaling makes it even stronger.** Across all settings, the RL-trained model consistently outperforms the corresponding base model. However, when equipped with multi-temperature sampling, the base model can match the *single-temperature* RL model under the same compute budget. At the same time, RL models themselves gain additional accuracy when combined with temperature scaling, indicating that RL does introduce new reasoning modes that a fixed temperature cannot reliably activate. (This is expected, since the verifier reduces uncertainty.) We view RL and temperature scaling as complementary rather than competing: RL-based frameworks such as TTRL [6] can incorporate temperature scaling during test-time exploration, and non-RL optimization frameworks such as Reasoning with Sampling [7] can likewise integrate temperature scaling as an additional axis of compute to further improve reasoning performance.
> > >
> > > ---
> > >
> > > ### [W4] Potential Cherry-Picking in Figure 3d
> > >
> > > **Reviewer comment:**
> > > *"It is unclear how the specific value of K (e.g., K used in Figure 3d) was chosen for Pass@K comparison. Increasing K might let single-T sampling eventually surpass the RL model. The comparison feels selective and may not show the full picture."*
> > >
> > > ---
> > >
> > > **Author response:**
> > >
> > > Thank you for raising this concern. The temperature $T=1.4$ follows the official recommendation of the Polaris release. We agree that using a single $(T,K)$ pair may introduce ambiguity, and reviewer `xcmB` raised a similar point. To address this, our expanded experiments in Table 1 and Table 3 use a *fixed token budget* for all methods, ensuring strictly fair comparisons across single-temperature, multi-temperature, self-consistency, verifier-assisted, RL, and thinking models. Under equal compute, we show that temperature scaling enables the base model to match or even exceed the *single-temperature* RL model. However, when RL models themselves are combined with temperature scaling, their achievable upper bound remains higher. Please refer to our responses in [W1] and [W3] for detailed results.

---

> ### Author Response · Authors · 2025-11-27
> **Rebuttal to Reviewer zrJ8 - Part 4**
>
> ### [Q1] Connection to Dynamic Temperature-Sampling Literature
>
> **Reviewer question:**
> *"There is existing work on models learning to dynamically adjust temperature during generation. Could the authors discuss this line of research and explain its connection to the current findings?"*
>
> ---
>
> **Author response:**
>
> Thank you for raising this question. Existing work on dynamic temperature sampling is largely **optimization-based**: the goal is to infer a single "best" temperature for the entire dataset or for each question. For example, prior work [8] estimates the optimal temperature from dataset-level entropy, which still results in one global temperature; as we show, restricting to a single temperature can severely limit performance on hard problems. Other approaches [9] train an adapter to predict an optimal temperature, yet in the evaluation such methods predict the same temperature (0.0) for MATH500, and clearly, this is suboptimal. In short, these methods attempt to **optimize a single temperature**, while our results indicate that such a notion of "best" temperature may not exist.
>
> Our findings suggest that different problems favor different temperatures because temperature reshapes the autoregressive trajectory distribution in problem-specific ways. Instead of optimizing for a single value, we therefore adopt a **scaling-based** approach: using a set of temperatures that jointly cover the union of solvable problems. With a verifier, easy problems exit immediately at the first temperature that produces a correct trace, which drastically reduces the effective cost of temperature scaling. As shown in Tables 1 and 3, under an equal token budget, multi-temperature sampling consistently outperforms single-temperature approaches.
>
> ---
>
> ### References
>
> 1. Liu et al., Can 1B LLM Surpass 405B LLM? Rethinking Compute-Optimal Test-Time Scaling, Reasoning and Planning for LLMs @ ICLR 2025
> 2. Xue et al., Dynamic Voting for Efficient Reasoning in Large Language Models, EMNLP Findings 2023.
> 3. Zhang et al., OpenPRM: Building Open-domain Process-based Reward Models with Preference Trees, ICLR 2025
> 4. Lee et al., Rethinking Reward Models for Multi-Domain Test-Time Scaling, Arxiv 2025
> 5. Yue et al., Does Reinforcement Learning Really Incentivize Reasoning Capacity in LLMs Beyond the Base Model, NeurIPS 2025
> 6. Zuo et al., TTRL: Test-Time Reinforcement Learning, NeurIPS 2025
> 7. Karan et al., Reasoning with Sampling: Your Base Model is Smarter Than You Think, Arxiv 2025
> 8. Du et al., Optimizing Temperature for Language Models with Multi-Sample Inference, ICML 2025
> 9. Dhuliawala et al., Adaptive Decoding via Latent Preference Optimization, Arxiv 2024
>
>
>
>
> ---
>
> We sincerely thank Reviewer `zrJ8` again for the detailed and constructive comments. We hope that our clarifications and revisions address all concerns and help strengthen the paper.

---

### Official Review · Reviewer_xcmB · 2025-11-07

**Soundness:** 3
**Presentation:** 2
**Contribution:** 2
**Rating:** 4
**Confidence:** 4

**Summary:**

This paper focuses on enhancing the reasoning performance of large language models through temperature scaling. The authors propose an innovative method that adjusts the temperature parameter during the testing phase to expand the model's reasoning capabilities. The study demonstrates that a multi-temperature strategy significantly outperforms a single-temperature strategy. This approach can achieve performance comparable to models trained with reinforcement learning, solely through appropriate temperature configuration during testing, without additional training. This method is both innovative and practical, warranting further exploration and discussion.

**Strengths:**

Introduces an innovative temperature scaling mechanism during the reasoning process.
The experimental design is sound, validating the effectiveness of the method across multiple datasets without increasing training burden.
Provides a detailed entropy analysis revealing the theoretical mechanism behind temperature scaling.

**Weaknesses:**

Terms like "reasoning boundary" and "upper bound" are ambiguously defined; the measurement criteria for the budget (such as the number of tokens, decoding steps, etc.) are not clearly specified, which may lead to unfair comparisons.
Insufficient comparison with strong baselines. Lacks rigorous comparisons under equivalent reasoning budgets with methods like single high-temperature self-consistency, multi-temperature grid + voting, best-of-N/majority voting, strong verifier-assisted methods, and RL models.
Key methodological details are incomplete. Lacks principles for selecting the temperature set, adaptive budget allocation, detailed voting mechanism, and sensitivity analysis for early exit strategy

**Questions:**

It is recommended to rigorously define terms such as "reasoning boundary" and "upper bound," and to clearly specify the metrics used for budgeting (e.g., total number of generated tokens, number of decoding steps), ensuring consistency across all comparative experiments.

---

> ### Author Response · Authors · 2025-11-27
> **Rebuttal to Reviewer xcmB - Part 1**
>
> We deeply appreciate Reviewer `xcmB`'s comprehensive and constructive feedback. In the following, we will address the concerns you raised.
>
> ---
>
> ### [W1Q1] Clarification of Terms and Definition of Budget Metrics
>
> **Reviewer comment:**
> *"Terms like 'reasoning boundary' and 'upper bound' are ambiguously defined; the measurement criteria for the budget (such as the number of tokens, decoding steps, etc.) are not clearly specified, which may lead to unfair comparisons."*
>
> **Reviewer question:**
> *"It is recommended to rigorously define terms such as 'reasoning boundary' and 'upper bound,' and to clearly specify the metrics used for budgeting (e.g., total number of generated tokens, number of decoding steps), ensuring consistency across all comparative experiments."*
>
> ---
>
> **Author response:**
>
> **Part 1. Clarification of Terms**
>
> Thank you for pointing out that the concepts of *"reasoning boundary"* and *"upper bound"* were not rigorously defined in the submission. We agree with this concern, and we now provide precise definitions:
>
> (1) **Solvable Set.** For a question $q$, a sampling temperature $T$, and a test-time budget $K$, we say that $q$ is *solvable* under $(T,K)$ if the success probability exceeds a small threshold $\epsilon>0$ (e.g., at least one verified correct trace among $K$ samples). Formally,
> \begin{align*}
> \mathcal{S}(T,K)=\lbrace q:\mathrm{Pass@K}(q;T)\ge\epsilon \rbrace.
> \end{align*}
>
> (2) **Single-Temperature Reasoning Boundary.** The *reasoning boundary* at a temperature $T$ is the set of all problems solvable under any allowed test-time budget (up to the maximum $K_{\max}$):
> $$
> \mathcal{B}(T)=\bigcup_{K\le K_{\max}}\mathcal{S}(T,K).
> $$
>
> (3) **Upper Bound.** The *upper bound* of a decoding configuration is the size of its reasoning boundary. When sampling across multiple temperatures, the expanded boundary is
> $$
> \mathcal{B}_ {\text{multi}}=\bigcup_{T\in\mathcal{T}}\mathcal{B}(T),
> $$
> representing the maximal set of problems the model can solve when exploring multiple temperatures.
>
> These definitions formalize our use of "reasoning boundary" and "upper bound," and match what is measured in our experiments: single-temperature TTS explores only $\mathcal{B}(T)$, while temperature scaling expands it to $\mathcal{B}_{\mathrm{multi}}$.
>
> **Part 2. Definition of Budget Metrics**
> The original submission primarily used the *number of sampled reasoning traces* (i.e., the sampling count $K$) as the test-time budget metric, and each trace corresponds to a full decoding trajectory. We agree with the reviewer that differences in reasoning-trace length may introduce ambiguity when comparing across methods.
>
> To remove this ambiguity, we now adopt a unified and model-agnostic budget metric in the experiment comparison part: **the cumulative number of generated tokens per question** (denoted as cumulative M tokens/question). This metric accounts for the total autoregressive decoding steps consumed across all sampled traces and ensures fair comparison across temperatures, decoding strategies, and methods with different trace lengths.
>
> ---
>
> ### [W2] Comparison With Stronger Baselines and Other Models
>
> **Reviewer comment:**
> *"Insufficient comparison with strong baselines. Lacks rigorous comparisons under equivalent reasoning budgets (e.g., single high-T self-consistency, multi-temperature grid + voting, BoN/majority vote, strong verifier-assisted methods, RL models)."*
>
> ---
>
> **Author response:**
>
> Thank you for pointing out that our comparison did not sufficiently cover stronger baselines and alternative decoding strategies. While the original submission focused primarily on establishing an upper bound for TTS, prior work [1] has shown that compute-efficient TTS is equally crucial for a fair and meaningful evaluation. We appreciate this suggestion, and we expand the experimental coverage to include stronger baselines, alternative decoding schemes, and models with RL or complex reasoning capabilities to fully address your concern.
>
> **Part 1. Comparisons with Other Decoding Methods Under Equivalent Budgets**
>
> To provide a more complete and fair comparison under token budgets, we evaluate seven different decoding configurations:
>
> 1. **Single-temperature (high-temperature T=1.2)**
> 2. **Multi-temperature (uniform grid, T ranging from 0.1 to 1.2)**
> 3. **Single-temperature + Self-Consistency (SC / majority voting)**
> 4. **Multi-temperature + SC**
> 5. **Single-temperature + Verifier**
> 6. **Multi-temperature + Verifier**
> 7. **Multi-temperature + Ours**
>
> Here, SC denotes a self-consistency style early-stop rule that terminates decoding once agreement is reached [2]. Verifier refers to the strongest oracle verifier: as soon as a correct answer is verified, the method stops generating further traces for that question. This isolates model-side effects from verifier quality [3,4] and guarantees fair comparison.

---

> ### Author Response · Authors · 2025-11-27
> **Rebuttal to Reviewer xcmB - Part 2**
>
> To extend the evaluation beyond the Qwen family, we also include the open-source SOTA models gpt-oss-20B and gpt-oss-120B.
>
> All models are evaluated under a fixed compute budget of 0.1M generated tokens per question. The resulting performance comparison is summarized in the following table.
>
> **Table 1: Performance of various decoding strategies on AIME 2025, evaluated under the same compute budget.**
> | **Model**               | **Single-temp** | **Multi-temp** | **Single-temp + SC** | **Multi-temp + SC** | **Multi-temp + Ours** | **Single-temp + Verifier** | **Multi-temp + Verifier** |
> |-------------------------|------------------|------------------|------------------------|------------------------|--------------------------|------------------------------|-----------------------------|
> | **Qwen3-4B**            | 43.41           | 45.94           | 43.41                 | 45.94                 | **47.49**                     | 45.85                       | 49.34                      |
> | **Qwen3-8B**            | 39.81           | 43.17           | 39.84                 | 43.21                 | **45.00**                     | 41.50                       | 45.40                      |
> | **gpt-oss-20B**         | 83.27           | 87.17           | 83.27                 | 87.17                 | **87.17**                     | 84.59                       | 90.59                      |
> | **gpt-oss-120B**        | 92.92           | 93.15           | 92.94                 | 93.17                 | **93.23**                     | 93.33                       | 93.58                      |
>
> **Multi-temperature sampling helps.** On AIME 2025, even the plain multi-temperature strategy (simply splitting the same budget across several temperatures) outperforms the single-temperature setting. This means different temperatures indeed solve different questions, and by scaling across temperatures we lose nothing while gaining better performance under the same budget.
>
> **Our early exit method is stronger than SC, and multi-temperature sampling remains better when we have a verifier.** When no verifier is available, our method outperforms SC. When a strong verifier is available, decoding stops immediately once a correct answer is verified, saving substantial compute. Multi-temperature sampling still achieves higher accuracy. The gap between single-temp and multi-temp varies: sometimes small, sometimes very large. This is because we cannot reliably know whether the chosen single temperature is the "good" one for a given model or dataset, which is why temperature scaling is needed.
>
> **Part 2. Temperature Scaling for RL and Thinking Models**
>
> Here we incorporate two models: Polaris-4B (RL-trained) and Qwen3-4B-Thinking (reasoning mode). These models naturally produce longer and more structured reasoning traces, making them a good testbed for whether temperature scaling continues to help under stronger reasoning capabilities. The results are shown in Table 2.
>
> **Table 2: AIME 2025 results for Polaris-4B and Qwen3-4B-Thinking.**
> | **Model**               | **Single-temp** | **Multi-temp** | **Single-temp + SC** | **Multi-temp + SC** | **Multi-temp + Ours** | **Single-temp + Verifier** | **Multi-temp + Verifier** |
> |-------------------------|------------------|------------------|------------------------|------------------------|--------------------------|------------------------------|-----------------------------|
> | **Polaris-4B**          | 86.67             | 89.80             | 86.67                   | 90.67                   | **91.13**|86.67                     | 96.67                                                 |
> | **Qwen3-4B-Thinking**   | 86.67             | 87.65             | 86.67                   | 87.88                   | **88.33**                     | 86.67                         | 90.00                        |
>
> **Temperature scaling also improves RL and thinking models.** Under the same compute budget, multi-temperature sampling outperforms single-temperature sampling, and the gap becomes large under a strong verifier (e.g., Polaris-4B improves by 10 points). This indicates that single-temperature sampling fails to explore many valid reasoning paths, while temperature scaling recovers that missing capability under same budget.

---

> ### Author Response · Authors · 2025-11-27
> **Rebuttal to Reviewer xcmB - Part 3**
>
> **RL is strong, and temperature scaling makes it even stronger.**  Prior work [5] suggested that RL-trained models may not provide fundamentally new capabilities, since a base model with extensive test-time sampling can match or surpass them. Our results give a more nuanced picture: Polaris-4B gains additional accuracy when combined with temperature scaling, indicating that RL *does* introduce new reasoning modes that a single temperature cannot reliably activate. (This is expected, since the verifier reduces uncertainty.) At the same time, a base model equipped with temperature scaling can exceed the single-temperature RL model, suggesting complementary strengths. We view RL and temperature scaling as compatible rather than competing, frameworks such as TTRL [6] and Reasoning with Sampling [7] can combine both to further enhance reasoning performance.
>
>
> ---
>
> ### [W3] Missing Methodological Details
>
> **Reviewer comment:**
> *"Key methodological details are incomplete. Lacks principles for selecting the temperature set, adaptive budget allocation, detailed voting mechanism."*
>
> ---
>
> **Author response:**
>
> Thank you for pointing out that the methodological details were not sufficiently described in the original submission. Due to space constraints, our explanations for temperature selection and early-exit design were brief, which may have caused confusion. In the revised version, we will incorporate following contents.
>
> **Part 1. Principled Selection of A Temperature Support Set**
> Given a grid of candidate temperatures
> \begin{align*}
> \mathcal{T}_ {\text{grid}} = \lbrace 0.0, 0.1, \ldots, 1.2 \rbrace
> \end{align*}
> and a fixed budget $K$, we first compute for each temperature $T \in \mathcal{T}_ {\text{grid}}$ its solvable set
> \begin{align*}
> \mathcal{S}(T) = \lbrace q : \mathrm{Pass@K}(q;T) \ge \epsilon \rbrace,
> \end{align*}
> using the same definition as in [W1Q1]. Let
> \begin{align*}
> \mathcal{U} = \bigcup_{T \in \mathcal{T}_ {\text{grid}}} \mathcal{S}(T)
> \end{align*}
> be the full union over all temperatures. Our goal is to find a small *support set* of temperatures $\mathcal{T}_ {\text{sup}} \subseteq \mathcal{T}_ {\text{grid}}$ such that
> \begin{align*}
> \Big|\bigcup_{T \in \mathcal{T}_ {\text{sup}}} \mathcal{S}(T)\Big|
> \ge \rho  |\mathcal{U}|,
> \end{align*}
> for a target coverage $\rho$ (e.g., $\rho = 0.99$). This is exactly a set cover problem, which we solve with the standard greedy algorithm: starting from $\mathcal{T}_ {\text{sup}} = \emptyset$, at each step we add the temperature
> \begin{align*}
> T^* = \arg\max_{T \in \mathcal{T}_ {\text{grid}} \setminus \mathcal{T}_ {\text{sup}}}
> \Big|\mathcal{S}(T) \setminus \bigcup_{T' \in \mathcal{T}_ {\text{sup}}} \mathcal{S}(T')\Big|,
> \end{align*}
> and stop when the coverage constraint above is met. Applying this procedure across our models and datasets, we obtain a support set $[0.4, 1.2]$ that recovers almost all solvable problems. This is the temperature set used in our multi-temperature experiments.
>
> **Part 2. Early-Exit Mechanism**
> For a given temperature $T$ and a fixed per-temperature budget $K_T$, decoding produces a sequence of candidate answers $\lbrace a_1, a_2, \dots \rbrace$. Our early-exit rule monitors the vote distribution
> $$
> p_t(y) = \frac{1}{t}\sum_{i=1}^t \mathbf{1}[a_i = y],
> $$
> and terminates decoding at the first step $t < K_T$ where either of the following conditions holds:
>
> (1) **Intra-temperature agreement:**
>    $$\max_y p_t(y) \ge \tau_{\text{intra}},$$
>    indicating that the samples at temperature $T$ have converged.
>
> (2) **Cross-temperature agreement:** if the current temperature $T$ is not the first one in the schedule, we additionally maintain the running majority vote across all temperatures processed so far. Let
> \begin{align*}
> p_t^{\le T}(y) = \text{normalized vote count across all visited temperatures},
> \end{align*}
> then we exit early if
> \begin{align*}
> \max_y p_{t}^{\le T}(y) \ge  \tau_{\text{cross}}.
> \end{align*}
>
> If neither condition is triggered, decoding continues until either $K_T$ samples are consumed or a verifier (when available) confirms correctness and forces immediate termination. The thresholds $\tau_{\text{intra}}$ and $\tau_{\text{cross}}$ are kept constant across all models and datasets; in the following [W4] part, we will include a sensitivity analysis showing the method is robust across a wide range of values.
>
> This mechanism generalizes SC by enabling consensus checks not only *within* a single temperature, but also *across multiple temperatures*. It ensures compute is concentrated on harder questions while retaining the benefits of temperature scaling.

---

> ### Author Response · Authors · 2025-11-27
> **Rebuttal to Reviewer xcmB - Part 4**
>
> ### [W4] Need Sensitivity Analysis for Early Exit Strategy
>
> **Reviewer comment:**
> *"Lacks sensitivity analysis for early exit strategy."*
>
> ---
>
> **Author response:**
>
> Thank you for the suggestion. We sweep the intra-temperature threshold $\tau_{\text{intra}}$ with $\tau_{\text{cross}} = 1.0$. Accuracy (in %) is shown below:
>
> **Table 3: Sweeping $\tau_{\text{intra}}$, Qwen3-4B on AIME 2025.**
> | $\tau_{\text{intra}}$ | 0.0  | 0.1  | 0.2  | 0.3  | 0.4  | 0.5  | 0.6  | 0.7  | 0.8  | 0.9  | 1.0  |
> |------------------------|------|------|------|------|------|------|------|------|------|------|------|
> | Accuracy (%)           |45.50 |45.50 |46.94 |47.26 |47.17 |47.03 |47.62 |47.57 |47.57 |47.35 |47.31 |
>
> We also fix $\tau_{\text{intra}} = 0.8$ and sweep $\tau_{\text{cross}}$, and the resulting accuracy remains unchanged (47.57% for all tested values).
>
> **These results show that the early-exit mechanism is stable:** varying either threshold across a wide range barely changes accuracy. This confirms the method is not sensitive to hyperparameters and simply triggers once the vote distribution becomes confidently peaked.
>
> ---
>
> ### Connection to Previous Temperature-Sampling Literature
>
> Existing work on dynamic temperature sampling is largely **optimization-based**: the goal is to infer a single "best" temperature for the entire dataset or for each question. For example, prior work [8] estimates the optimal temperature from dataset-level entropy, which still results in one global temperature; as we show, restricting to a single temperature can severely limit performance on hard problems. Other approaches [9] train an adapter to predict an optimal temperature, yet in the evaluation such methods predict the same temperature (0.0) for MATH500, and clearly, this is suboptimal. In short, these methods attempt to **optimize a single temperature**, while our results indicate that such a notion of "best" temperature may not exist.
>
> Our findings suggest that different problems favor different temperatures because temperature reshapes the autoregressive trajectory distribution in problem-specific ways. Instead of optimizing for a single value, we therefore adopt a **scaling-based** approach: using a set of temperatures that jointly cover the union of solvable problems. With a verifier, easy problems exit immediately at the first temperature that produces a correct trace, which drastically reduces the effective cost of temperature scaling. As shown in Tables 1 and 3, under an equal token budget, multi-temperature sampling consistently outperforms single-temperature approaches.
>
> ---
>
> ### References
>
> 1. Liu et al., Can 1B LLM Surpass 405B LLM? Rethinking Compute-Optimal Test-Time Scaling, Reasoning and Planning for LLMs @ ICLR 2025
> 2. Xue et al., Dynamic Voting for Efficient Reasoning in Large Language Models, EMNLP Findings 2023.
> 3. Zhang et al., OpenPRM: Building Open-domain Process-based Reward Models with Preference Trees, ICLR 2025
> 4. Lee et al., Rethinking Reward Models for Multi-Domain Test-Time Scaling, Arxiv 2025
> 5. Yue et al., Does Reinforcement Learning Really Incentivize Reasoning Capacity in LLMs Beyond the Base Model, NeurIPS 2025
> 6. Zuo et al., TTRL: Test-Time Reinforcement Learning, NeurIPS 2025
> 7. Karan et al., Reasoning with Sampling: Your Base Model is Smarter Than You Think, Arxiv 2025
> 8. Du et al., Optimizing Temperature for Language Models with Multi-Sample Inference, ICML 2025
> 9. Dhuliawala et al., Adaptive Decoding via Latent Preference Optimization, Arxiv 2024
>
> ---
>
> We sincerely thank the reviewer `xcmB` again for their insightful comments. We hope our additions and clarifications address all raised concerns and contribute to a more rigorous and transparent presentation of our work.

---

### Author Response · Authors · 2025-12-03
**To Area Chair - Rebuttal Summary**

Dear Area Chair,

Thank you for overseeing the review process. To support your assessment, we summarize the paper's contributions, the key issues raised by the reviewers, and the improvements made in the rebuttal process.

---

### Background and Contributions

Test-time scaling (TTS) methods typically rely on a single sampling temperature, which restricts the exploration of a model's reasoning space. We find that different temperatures unlock different reasoning trajectories, particularly on hard problems. Based on this observation, we propose temperature scaling, which expands test-time compute along a new dimension and improves reasoning performance under the same computation budget.

Our contributions are:

1. **Scaling test-time compute to a new dimension.**  Varying temperature expands the model's reasoning boundary, revealing an orthogonal axis of test-time compute.

2. **Analyzing the dynamics of temperature scaling.** We show that easy questions are solvable across all temperatures, while hard questions require different temperatures.

3. **Efficient multi-temperature inference.** An early-exit voting strategy solves easy questions quickly while preserving the benefit of temperature scaling.

4. **Compute-equivalent gains.** Under the same cumulative token budget, temperature scaling outperforms single-temperature sampling.

---

### Reviewer Feedback and Improvements

**1. Need clearer definitions of key concepts and budget metrics** (xcmB)

We added definitions for solvable sets, reasoning boundaries, and upper bounds, and added comparisons use the same cumulative token budget per question.

---

**2. Need stronger baselines and more diverse models under equal compute**  (xcmB, zrJ8, 3DQD, j3rw)

We added extensive equal-budget comparisons including:
- seven decoding schemes (single-T, multi-T, SC, SC+multi-T, our method, verifier single-T, verifier multi-T),
- large models (20B, 120B),
- RL-trained and Thinking models.

Results show that temperature scaling matches or surpasses single-temperature TTS under the same computation budget.

---

**3. Need methodological details (temperature selection, allocation, voting)** (xcmB, j3rw)

We added:
- a formal method description for constructing a minimal temperature support set,
- detailed descriptions of cross- and intra-temperature thresholds,
- full pseudo-code for the multi-temperature inference algorithm.

---

**4. Need sensitivity analysis and ablation studies** (xcmB, j3rw)

We included sweeps of intra-temperature thresholds (0.0–1.0), cross-temperature thresholds, and analysis showing stable performance and efficient early exit behavior.

---

**5. Need deeper explanation for why different temperatures solve different problems** (zrJ8, 3DQD)

We added an analysis showing how temperature rescales token-level logits, altering branching probabilities in the autoregressive process. Different questions have different branching structures, explaining the temperature-problem dependency.

---

**6. Need clearer connection to prior work on dynamic temperature selection** (zrJ8)

We added a discussion contrasting optimization-based approaches (predicting a single temperature) with our scaling-based approach. Prior methods collapse to one temperature, while our findings show that no single temperature is optimal for hard problems, motivating temperature scaling.

---

We appreciate the reviewers' and your time, effort, and thoughtful feedback throughout this process. We believe the revisions substantially strengthen the clarity, rigor, and completeness of the paper.

Sincerely,

The Authors

---

### Meta-Review · Area_Chair_N4J7 · 2025-12-31

**Summary:**

This paper investigates the role of temperature sampling in test-time scaling. While prior work shows that increasing the number of samples improves accuracy, the manuscript shows that performance eventually plateaus and, thus, using different temperatures provides a significant benefit: the different samples of temperature appear to solve different problems. The proposed multi-temperature voting strategy outperforms various baselines for LLMs in the Qwen family.

Overall, all reviewers have praised the innovative temperature scaling mechanism, the sound experimental design and the easy-to-follow narrative. However, all the 4 reviewers (despite a slight difference in scores) point to similar issues, which can be summarized as follows:

(1) Experiments are conducted only on the Qwen3 series models and restricted to up to 8B parameters.

(2) The paper lacks a theoretical framework to explain the findings.

(3) The baseline comparison is deemed insufficient.

(4) Insufficient ablation on hyperparameters.

(5) Several terms are not rigorously defined.

To address these issues, the authors have provided an extensive rebuttal, with detailed explanations and several new experiments. However, in my opinion, the rebuttal does not fully solve the issues above and, to do that, an additional round of reviewing/revisions is required. Therefore, I recommend a rejection at this stage, encouraging the authors to resubmit a revision to a future venue.

**Reviewer Concerns:**

I have outlined above 5 main issues that are recurring in the 4 reviews. In my opinion, the rebuttal of the authors conclusively solves (4) and (5). The lack of theoretical framework (2) is somewhat hard to solve but given the experimental nature of the paper, I do not think this is a deal-breaker. My concerns are mostly about (1) and (3); the novel experiments proposed by the authors are not a minor revision and require another round of reviewing to be properly assessed. For example, the gains for gpt-oss-120B appear essentially negligible and could very well be in the confidence interval of the experiment (not reported).

**Reviewer Scores:**

While it is possible that some reviewer could have changed score, I find it unlikely that a consensus would have been reached towards accepting the paper in its current state.

---

### Decision · Program_Chairs · 2026-01-26

Reject